

# Mamba Goes HoME:
# Hierarchical Soft Mixture-of-Experts
# for 3D Medical Image Segmentation

**Szymon Płotka**[1,2*]  **Gizem Mert**[3]  **Maciej Chrabaszcz**[4,5]
**Ewa Szczurek**[1,3]  **Arkadiusz Sitek**[6,7]

[1] Faculty of Mathematics, Informatics, and Mechanics, University of Warsaw, Poland
[2] Faculty of Mathematics and Computer Science, Jagiellonian University, Poland
[3] Institute of AI for Health, Helmholtz Munich, Germany
[4] Faculty of Electronics and Information Technology, Warsaw University of Technology, Poland
[5] NASK - National Research Institute, Poland
[6] Faculty of Radiology, Massachusetts General Hospital, USA
[7] Department of Radiology, Harvard Medical School, USA

## Abstract

In recent years, artificial intelligence has significantly advanced medical image segmentation. Nonetheless, challenges remain, including efficient 3D medical image processing across diverse modalities and handling data variability. In this work, we introduce Hierarchical Soft Mixture-of-Experts (HoME), a two-level token-routing layer for efficient long-context modeling, specifically designed for 3D medical image segmentation. Built on the Mamba Selective State-Space Model (SSM) backbone, HoME enhances sequential modeling through adaptive expert routing. In the first level, a Soft Mixture-of-Experts (SMoE) layer partitions input sequences into local groups, routing tokens to specialized per-group experts for localized feature extraction. The second level aggregates these outputs through a global SMoE layer, enabling cross-group information fusion and global context refinement. This hierarchical design, combining local expert routing with global expert refinement, enhances generalizability and segmentation performance, surpassing state-of-the-art results across datasets from the three most widely used 3D medical imaging modalities and varying data qualities. The code is publicly available at github.com/gmum/MambaHoME.

## 1 Introduction

Three-dimensional (3D) medical image segmentation lies at the core of computer-aided diagnosis, image-guided interventions, and treatment planning across modalities such as Computed Tomography (CT) [51, 12, 44], Magnetic Resonance Imaging (MRI) [33, 48, 21], and Ultrasound (US) [60, 24, 6]. A key characteristic of medical imaging data is its hierarchical structure: local patterns, such as tumor lesions, are embedded within larger anatomical structures like organs, which themselves follow a consistent global arrangement [20]. We hypothesize that *models capable of capturing these local-to-global spatial hierarchies in 3D medical data can enhance segmentation performance and yield latent representations that generalize effectively across diverse imaging modalities*.

Convolutional Neural Networks (CNNs) provide local feature extraction with linear complexity in the number of input pixels, but their limited receptive fields hinder their ability to capture global

---

*Correspondence to: Szymon Płotka (SZYMON.PLOTKA@UJ.EDU.PL)

39th Conference on Neural Information Processing Systems (NeurIPS 2025).

spatial patterns [42, 57, 22, 52, 45]. In contrast, Vision Transformers (ViTs) [16, 5, 31] leverage global attention mechanisms to model long-range dependencies. However, their quadratic complexity in the number of tokens makes them computationally costly for high-resolution 3D data, posing significant challenges for scalability. Moreover, while many ViT-based models [53, 18, 38, 30, 27, 47] incorporate multi-scale feature extraction and attention mechanisms, they still struggle to effectively aggregate fine-to-coarse semantic information, particularly in dense prediction tasks such as volumetric segmentation. Although these models demonstrate improved performance, they do not explicitly model global and local spatial patterns and their mutual arrangement. Modeling the transition from local to global patterns requires processing long sequences and capturing long-range dependencies, imposing prohibitive memory and computational demands on current architectures [9].

Recently, Selective State-Space Models (SSMs), such as Mamba [19, 15], have emerged as efficient alternatives to ViTs by offering linear complexity in the number of tokens to capture long-range dependencies, including in 3D medical imaging [54, 46, 29, 11, 50]. Although SSMs effectively capture global context with lower computational cost than attention-based methods, they are not inherently designed to adaptively handle diverse local patterns in medical data. Efficient management of such local patterns in complex, multi-scale data while maintaining scalability is achieved through Mixture-of-Experts (MoE) frameworks, which dynamically route features to specialized subnetworks. MoE-based methods have gained prominence across domains such as language modeling [4, 56, 26], vision tasks [39, 59, 41, 14, 17], multimodal learning [34, 8, 55], and medical applications [25, 13, 49, 38]. However, combining the global efficiency of SSMs with the localized adaptability of MoE remains largely unexplored, particularly in 3D medical imaging, where balancing efficiency and generalization across multi-modal datasets under resource constraints is paramount.

To address these challenges, we introduce the first-in-class model that smoothly integrates Mamba with hierarchical Soft MoE in a multi-stage network for local-to-global pattern modeling and 3D image segmentation with competitive memory and compute efficiency, while maintaining state-of-the-art segmentation performance. Specifically, our contributions are as follows:

1. We introduce **H**ierarchical **S**oft **M**ixture-of-**E**xperts (**HoME**), a two-level, token-routing MoE layer for efficient capture of local-to-global pattern hierarchies, where tokens are grouped and routed to local experts in the first SMoE level, then aggregated and passed via a global SMoE in the second level,

2. We design a unified architectural block that integrates Mamba's SSMs with HoME, combining memory-efficient long-sequence processing with hierarchical expert routing,

3. We embed the above novel solutions into a multi-stage U-shaped architecture, called **Mamba-HoME**, specifically designed for 3D medical image segmentation, where the integration of hierarchical memory and selective state-space modeling enhances contextual representation across spatial and depth dimensions.

Through comprehensive experiments on four publicly available datasets, including PANORAMA [2], AMOS [23], FeTA 2022 [37], and MVSeg [10] as well as one in-house CT dataset, we demonstrate that Mamba-HoME outperforms current state-of-the-art methods in both segmentation accuracy and computational efficiency, while generalizing effectively across three major 3D medical imaging modalities: CT, MRI, and US.

## 2 Methodology

### 2.1 Preliminaries

**Selective State-Space Models (Mamba).** Our network builds on the Mamba layer [19], designed for long-sequence modeling with linear computational complexity. Unlike Transformer-based architectures, which exhibit quadratic complexity with respect to sequence length, Mamba achieves linear-time processing through a continuous-time recurrent formulation with input-dependent parameters. Given a feature sequence $z = \{z_t\}_{t=1}^N \in \mathbb{R}^{B \times N \times d}$, the hidden state $h_t$ and output $y_t$ are updated as follows:

$$h_t = A(z_t)h_{t-1} + B(z_t)z_t, \qquad y_t = C(z_t)h_t, \tag{1}$$

where $A(\cdot), B(\cdot), C(\cdot)$ are learnable input-dependent linear mappings. This formulation allows Mamba to capture both short- and long-range dependencies with a memory and computational

complexity of $\mathcal{O}(Nd)$. In our architecture, the Mamba layer serves as a backbone for volumetric sequence modeling, efficiently capturing spatial dependencies across 3D data inputs.

**Soft Mixture-of-Experts (SMoE).** SMoE framework [39] introduces a modular computation strategy where each input token is dynamically routed to multiple experts, and their outputs are combined using soft routing weights. Given an input tensor $x \in \mathbb{R}^{B \times N \times d}$ and $E$ experts, a gating network computes routing probabilities $p_{b,n,e}$ for each expert $e$ and token $x_{b,n}$:

$$y_{b,n} = \sum_{e=1}^{E} p_{b,n,e} \cdot \text{FFN}_e(x_{b,n}), \qquad y_{b,n} \in \mathbb{R}^d, \tag{2}$$

where $\text{FFN}_e$ denotes the $e$-th expert network and $\sum_{e=1}^{E} p_{b,n,e} = 1$ ensures a valid soft assignment. Each expert is typically a small feed-forward network, and the gating network is a learned per-token function (e.g., an MLP) that assigns weights dynamically. While SMoE improves model capacity and modularity, global routing over all tokens incurs a computational cost of $\mathcal{O}(NdE)$, which becomes prohibitive for high-resolution 3D inputs where $N$ can reach millions. Moreover, SMoE lacks spatial locality and hierarchical aggregation, both of which are important for structured volumetric data.

**Notation.** At encoder stage $i \in \{1, \dots, I\}$, the input feature sequence is $x \in \mathbb{R}^{B \times N_i \times d}$, where $B$ is the batch size, $N_i$ is the sequence length, and $d$ is the feature dimension. The sequence is divided into $G_i = \lceil N_i / K_i \rceil$ groups of up to $K_i$ tokens each. We define $E_i$ as the number of experts per level and $S_i$ as the number of slots per expert. The total number of expert-slot pairs (total slots) is $M_i = E_i \cdot S_i$, and the learnable slot embeddings are $E_{\text{slots}}^{(i)} \in \mathbb{R}^{M_i \times d}$. Tokens are assigned to slots using weights $A_{b,g,k,m}$. The first-level experts $E_{1,i}$ process the grouped slots to produce outputs $y^{(1)}$, which are then refined by a set of second-level experts $E_{2,i}$ to produce the final outputs $y^{(2)}$.

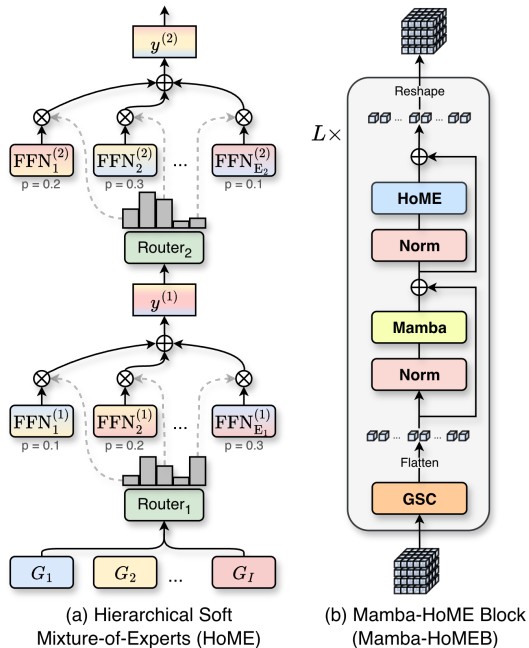

(a) Hierarchical Soft Mixture-of-Experts (HoME)

(b) Mamba-HoME Block (Mamba-HoMEB)

Figure 1: An overview of the HoME layer and Mamba-HoME Block design. (a) The HoME layer processes $G_i$ groups of $K_i$ tokens through a hierarchical soft routing. Tokens are soft-assigned to slots and projected by $E_1$ local experts ($\text{FFN}^{(1)}$) for intra-group feature extraction ($y^{(1)}$). Subsequently, $E_2$ global experts ($\text{FFN}^{(2)}$) refine these representations to enable inter-group communication and global context integration ($y^{(2)}$). (b) Mamba-HoMEB sequentially integrates Gated Spatial Convolution (GSC) for local spatial priors, which is flattened into a 1D sequence for long-range modeling within the Mamba layer. Following specialized refinement by the HoME layer, the output is reshaped to restore its original dimensionality. Dynamic Tanh ensures stable normalization and efficient computation across the architecture.

## 2.2 Hierarchical Soft Mixture-of-Experts (HoME)

We introduce the HoME layer (see Figure 1(a)), which enhances feature processing through a hierarchical two-level structure. HoME extends the SMoE concept with a hierarchical two-level routing structure that processes grouped tokens locally and enables inter-group information exchange efficiently. It comprises three key steps: (1) grouped slot assignment for token processing, (2) first-level MoE processing for local feature extraction, and (3) second-level MoE refinement for global feature extraction, allowing inter-group communication.

**Grouped Slot Assignment.** In hierarchical vision encoders, particularly for dense 3D inputs (e.g., volumetric data), the sequence length is highest in early stages due to large spatial resolutions. Global expert routing on these long sequences causes high computational and memory costs, as each token is compared to all expert slots, yielding a complexity of $\mathcal{O}(N_i \cdot M_i)$, where $N_i$ is the sequence length at stage $i$ and $M_i$ is the total number of expert slots. To address this scalability bottleneck, we introduce Grouped Slot Assignment, a locality-aware routing mechanism that divides the input

sequence into groups and performs soft assignment independently per group. We define the group size as $K_i = K_1 \cdot \rho^{i-1}$ ($0 < \rho < 1$), and zero-pad sequences to $N_i' = G_i \cdot K_i$, giving $\hat{x} \in \mathbb{R}^{B \times N_i' \times d}$. Each group $x_g \in \mathbb{R}^{B \times K_i \times d}$ is routed independently. Assignment logits are computed via a dot product between each token and learnable expert-specific slot embeddings $E_{\text{slots}}^{(i)} \in \mathbb{R}^{E_i \times S_i \times d}$. Let $e_1 \in \{1, \ldots, E_{1,i}\}$ and $s \in \{1, \ldots, S_i\}$. The logits are as follows:

$$S_{b,g,k,e_1,s} = \sum_{j=1}^{d} x_{b,g,k,j} \cdot E_{e_1,s,j}^{(i)}, \quad S \in \mathbb{R}^{B \times G_i \times K_i \times E_{1,i} \times S_i}. \tag{3}$$

An optional binary mask $M \in \{0,1\}^{B \times N_i}$, indicating valid (unpadded) tokens, is extended to $\hat{M} \in \{0,1\}^{B \times N_i'}$ after padding. To prevent padded tokens from affecting expert assignment, the logits for invalid tokens are masked by setting:

$$S_{b,g,k,e_1,s} = \begin{cases} S_{b,g,k,e_1,s}, & \text{if } \hat{M}_{b,\,gK_i+k} = 1, \\ -\infty, & \text{otherwise.} \end{cases} \tag{4}$$

Expert-slot pairs are flattened ($m = (e-1)S_i + s$, $M_i = E_i \cdot S_i$), and normalized with softmax:

$$A_{b,g,k,m} = \frac{\exp(S_{b,g,k,m})}{\sum_{m'=1}^{M_i} \exp(S_{b,g,k,m'})}, \quad A \in \mathbb{R}^{B \times G_i \times K_i \times M_i}. \tag{5}$$

Each slot representation is computed as a weighted aggregation of tokens within its group:

$$\tilde{x}_{b,g,e_1,s,j} = \sum_{k=1}^{K_i} A_{b,g,k,m} \cdot x_{b,g,k,j}, \quad \tilde{x} \in \mathbb{R}^{B \times G_i \times E_{1,i} \times S_i \times d}. \tag{6}$$

By performing routing within groups, peak memory usage is reduced and locality is preserved, enabling efficient hierarchical token-to-expert assignment. While the total computational complexity remains $\mathcal{O}(N_i M_i d)$, the smaller group-wise computations allow scalable processing of long sequences while being memory-efficient.

**Hierarchical Expert Processing.** Let $\tilde{x}^\flat \in \mathbb{R}^{B \times G_i \times M_i \times d}$ denote the grouped slot representations after flattening the $(e_1, s)$ dimensions of $\tilde{x}$. The first level routes slots within each group to a subset of $E_{1,i}$ experts, promoting local specialization and feature refinement. This produces group-specific outputs while preserving slot structure. For each group $g \in \{1, \ldots, G_i\}$, the gating network computes routing weights for the $E_{1,i}$ experts ($\text{FFN}_1, \text{FFN}_2, \ldots, \text{FFN}_{E_{1,i}}$):

$$\text{Router}_1(\tilde{x}_{b,g}^\flat; \theta_{\text{gate}}^{(1)}) = \text{softmax}\left(\text{MLP}_1\left(\frac{1}{M_i} \sum_{m=1}^{M_i} \tilde{x}_{b,g,m}^\flat\right)\right), \tag{7}$$

where $\tilde{x}_{b,g}^\flat = \{\tilde{x}_{b,g,m}^\flat\}_{m=1}^{M_i} \in \mathbb{R}^{M_i \times d}$, $\theta_{\text{gate}}^{(1)}$ are gating parameters, and the softmax normalizes over the expert dimension, yielding $\text{Router}_1 \in \mathbb{R}^{E_i}$. Each expert, implemented as an $\text{FFN}_{e_1} : \mathbb{R}^{M_i \times d} \to \mathbb{R}^{M_i \times d}$, processes the group's slots independently. The output for expert $e_1 \in \{1, \ldots, E_{1,i}\}$ is

$$y_{b,g,e_1} = \left[\text{Router}_1(\tilde{x}_{b,g}^\flat; \theta_{\text{gate}}^{(1)})\right]_{e_1} \cdot \text{FFN}_{e_1}(\tilde{x}_{b,g}^\flat), \tag{8}$$

and the aggregated output is:

$$y_{b,g}^{(1)} = \sum_{e_1=1}^{E_{1,i}} y_{b,g,e_1}, \quad y^{(1)} \in \mathbb{R}^{B \times G_i \times M_i \times d}. \tag{9}$$

**Dynamic Slot Refinement.** The second level routes group slots to $E_{2,i}$ experts for global feature refinement. The tensor $y^{(1)}$ is transformed into $\tilde{y} \in \mathbb{R}^{B \times (G_i M_i) \times d}$ before being passed through the second-level experts. The gating network computes routing weights for $E_{2,i}$ experts ($\text{FFN}_1, \text{FFN}_2, \ldots, \text{FFN}_{E_{2,i}}$):

$$\text{Router}_2(\tilde{y}; \theta_{\text{gate}}^{(2)}) = \text{softmax}(\text{MLP}_2(\tilde{y})), \tag{10}$$

where $\theta_{\text{gate}}^{(2)}$ are the gating parameters, producing $\text{Router}_2 \in \mathbb{R}^{B \times (G_i M_i) \times E_{2,i}}$. Each second-level expert, implemented as $\text{FFN}_{e_2} : \mathbb{R}^{(G_i M_i) \times d} \to \mathbb{R}^{(G_i M_i) \times d}$, processes $\tilde{y}$. The output for expert $e_2 \in \{1, \ldots, E_{2,i}\}$ is:

$$y_{b,e_2} = \left[\text{Router}_2(\tilde{y}_b; \theta_{\text{gate}}^{(2)})\right]_{e_2} \cdot \text{FFN}_{e_2}(\tilde{y}_b), \tag{11}$$

and outputs are aggregated into:

$$y^{(2)} = \sum_{e_2=1}^{E_{2,i}} y_{b,e_2}, \qquad y^{(2)} \in \mathbb{R}^{B \times (G_i M_i) \times d}. \tag{12}$$

The final stage reconstructs a structured output using an attention-based combination to emphasize relevant slots and remove padding, yielding a compact, task-aligned representation. The tensor $y^{(2)}$ is reshaped to $\mathbb{R}^{B \times G_i \times M_i \times d}$. For batch element $b$, group $g$, and token $k \in \{1, \ldots, K_i\}$, the output is:

$$y_{b,g,k} = \sum_{m=1}^{M_i} W_{b,g,k,m} \cdot y_{b,g,m}^{(2)}, \tag{13}$$

where $y_{b,g,m}^{(2)} \in \mathbb{R}^d$, and $W_{b,g,k,m} \in [0,1]$ are attention weights satisfying $\sum_{m=1}^{M_i} W_{b,g,k,m} = 1$. After undoing padding, the final output is $\hat{y} \in \mathbb{R}^{B \times N_i \times d}$.

## 2.3 Mamba-HoME Block

Figure 1(b) provides an overview of the proposed Mamba-HoME Block (Mamba-HoMEB). The Mamba-HoMEB extends the Mamba layer by incorporating hierarchical downsampling, Gated Spatial Convolution (GSC) module [54], and a Hierarchical Soft Mixture-of-Experts (HoME) layer (see Section 2.2). To improve gradient stability and computational efficiency in SSMs, we use Dynamic Tanh (DyT) [58] normalization, defined as:

$$f_{\text{DyT}}(x) = w \cdot \tanh(\alpha \cdot x) + b, \tag{14}$$

where $x \in \mathbb{R}^{B \times d}$ is the input tensor, $w, b \in \mathbb{R}^d$ are learnable per-channel vector parameters, and $\alpha \in \mathbb{R}$ is a shared scalar. DyT applies a point-wise nonlinearity, leveraging the bounded nature of $\tanh$ to stabilize gradients. Unlike Layer Normalization [7], it avoids costly mean and variance computations, reducing overhead while maintaining $\mathcal{O}(Bd)$ complexity. DyT accelerates training and validation with performance on par with normalization-based alternatives.

Given a 3D input volume $x \in \mathbb{R}^{B \times C \times D \times H \times W}$, the initial feature map $x_1^0 \in \mathbb{R}^{B \times 48 \times \frac{D}{2} \times \frac{H}{2} \times \frac{W}{2}}$ is extracted by a stem layer. This feature map $x_1^0$ is then passed through each Mamba-HoMEB and its corresponding downsampling layers. For the $l$-th layer ($l \in \{0, 1, \ldots, L_i - 1\}$) within stage $i$, the representations for the $i$-th Mamba-HoMEB are given by:

$$x_i'^l = f_{\text{GSC}}(x_i^l), \qquad \bar{x}_i^l = f_{\text{Mamba}}(f_{\text{Norm}}(x_i'^l)) + x_i'^l, \qquad x_i^{l+1} = f_{\text{HoME}}(f_{\text{Norm}}(\bar{x}_i^l)) + \bar{x}_i^l. \tag{15}$$

where $f_{\text{GSC}}$ denotes Gated Spatial Convolution module, $f_{\text{Mamba}}$ the Mamba layer, $f_{\text{Norm}}$ corresponds to DyT normalization (see Eq. 14), and $f_{\text{HoME}}$ the HoME layer. After applying $f_{\text{GSC}}$, the feature map is flattened along the spatial dimensions into a sequence (length $N_i$) before being processed by $f_{\text{Mamba}}$. The output of the HoME layer is then reshaped back into the original volumetric form to yield $x_i^{l+1}$.

The HoME layer operates hierarchically at each encoder stage $i \in \{1, \ldots, I\}$, with the number of first-level experts denoted by $E_i$ and the group size (i.e., the number of tokens processed jointly within each local group) denoted by $K_i$. The number of experts $E_i$ increases monotonically with stage depth, reflecting increased specialization ($E_1 < E_2 < \cdots < E_I$), while the group size $K_i$ decreases, enabling progressively finer-grained processing ($K_1 > K_2 > \cdots > K_I$). In addition to the first-level experts, each stage employs a second-level expert set $E_{2,i}$, which scales proportionally with $E_i$, i.e., $E_{2,i} = 2E_i$. This second level facilitates global context integration across groups, enhancing inter-group communication. The HoME operation at each stage thus combines local expert routing with global feature aggregation (see Section 2.2).

## 2.4 The Mamba-HoME Architecture

Building upon SegMamba [54], we introduce a U-shaped encoder-decoder network, called **Mamba-HoME**, designed for 3D medical image segmentation. It leverages a Mamba-based encoder backbone to efficiently capture both long-range dependencies and local features. The model processes an input 3D volume $x \in \mathbb{R}^{B \times C \times D \times H \times W}$ and produces a segmentation mask $y \in \mathbb{R}^{B \times C' \times D \times H \times W}$, where $B$ is the batch size, $C$ is the number of input channels, $C'$ is the number of output classes, and $D, H, W$ are spatial dimensions.

The encoder begins with a stem layer that produces the initial feature map $x_1^0$ (see Section 2.3), followed by $I$ hierarchical stages. Each encoder stage $i \in \{1, \ldots, I\}$ applies a Mamba-HoMEB, producing intermediate representations $x_i^l$ for $l = 0, \ldots, L_i - 1$, where $L_i$ denotes the number of layers in the $i$-th Mamba-HoMEB, and concludes with a downsampling operation:

$$x_{i+1}^0 = \text{Downsample}_i(x_i^{L_i}), \tag{16}$$

where downsampling halves the spatial dimensions and doubles the channel depth: $D_{i+1} = \lfloor D_i/2 \rfloor$, $H_{i+1} = \lfloor H_i/2 \rfloor$, $W_{i+1} = \lfloor W_i/2 \rfloor$, and $C_{i+1} = 2C_i$. This process results in a sequence of encoder feature maps $\{x_1^{L_1}, x_2^{L_2}, \ldots, x_I^{L_I}\}$, with $x_I^{L_I}$ serving as the bottleneck representation.

The decoder mirrors the encoder with $I - 1$ upsampling stages. From the bottleneck $u_0 = x_I^{L_I}$, each decoder stage $j = 1, \ldots, I - 1$ fuses the corresponding encoder skip connection with the upsampled decoder features:

$$u_j = \text{UpBlock}_j\big(x_{I-j}^{L_{I-j}} \oplus \text{Up}(u_{j-1})\big), \tag{17}$$

where $x_{I-j}^{L_{I-j}} \in \mathbb{R}^{B \times C_{I-j} \times D_{I-j} \times H_{I-j} \times W_{I-j}}$ is the skip connection from the encoder stage $I - j$, and $\oplus$ denotes channel-wise concatenation. The upsampled spatial dimensions are $D_j = \lfloor D/2^{I-j} \rfloor$, $H_j = \lfloor H/2^{I-j} \rfloor$, and $W_j = \lfloor W/2^{I-j} \rfloor$. Each $\text{UpBlock}_j$ refines the combined features and maps them to the target resolution for the next decoding stage, and a final prediction layer converts $u_{I-1}$ to the segmentation mask $y \in \mathbb{R}^{B \times C' \times D \times H \times W}$.

The computational complexity of Mamba-HoME scales as $\mathcal{O}(BN_i d)$ for the Mamba layer and $\mathcal{O}\big(B\, G_i\, (E_i + E_{2,i})\, L_i\, d\big)$ for the HoME layer at stage $i$, where $N_i$ is the token sequence length, $d$ is the hidden dimension, $G_i$ is the number of groups, and $E_i + E_{2,i}$ reflects the total number of experts. Compared to Transformer-based models ($\mathcal{O}(BN_i^2 d)$), this linear scaling in $N_i$ ensures efficiency for large 3D volumes (e.g., $N_i \approx 10^6$ tokens), with HoME adding a modest overhead proportional to the expert count.

## 3 Experiments

In this section, we evaluate the performance of the proposed Mamba-HoME against state-of-the-art methods for 3D medical image segmentation (see Section 3.3). We perform an ablation study to understand the importance of different configurations and parameters of Mamba-HoME. Moreover, we compare the segmentation performance of Mamba-HoME trained from scratch with a version pre-trained using a supervised learning approach (see Section 3.4). Finally, in Section 3.5, we investigate how Mamba-HoME generalizes to new modalities. Throughout this section, the best and second-best performing results are **bolded** and underlined, respectively.

Table 1: Segmentation and efficiency performance on the PANORAMA and in-house test sets. PDAC and P denote pancreatic ductal adenocarcinoma and pancreas, respectively. Methods marked with (*) are initialized with pre-trained weights.

| Method | DSC (%) ↑ PDAC | DSC (%) ↑ P | mDSC (%) ↑ | mHD95 (mm) ↓ | Params (M) ↓ | GPU (G) ↓ | IS(‡) ↓ |
|---|---|---|---|---|---|---|---|
| VoCo-B (*) [53] | 40.5 | 86.3 | 71.6 | 15.9 | 53.0 | 17.1 | 1.5 |
| Hermes [18] | 48.2 | 87.8 | 75.1 | 10.4 | 44.5 | 17.4 | 1.9 |
| Swin SMT [38] | 49.4 | 87.0 | 75.0 | 10.1 | 170.8 | 15.8 | 1.3 |
| VSmTrans [30] | 50.3 | 87.2 | 75.4 | 9.5 | 47.6 | 11.1 | 1.7 |
| uC 3DU-Net [22] | 52.0 | 88.2 | 76.6 | 8.4 | **21.7** | 13.6 | **1.0** |
| Swin UNETR [47] | 46.3 | 87.4 | 74.2 | 10.6 | 72.8 | 17.1 | 1.5 |
| SegMamba [54] | 49.7 | **88.5** | 76.0 | 8.6 | 66.8 | **10.1** | 1.2 |
| SuPreM (*) [28] | 51.7 | 88.3 | 76.6 | 4.7 | 62.2 | 17.1 | 1.5 |
| **Mamba-HoME** | 54.8 | 88.3 | 77.5 | 4.8 | 170.1 | 11.1 | 1.5 |
| **Mamba-HoME (*)** | 56.7 | **88.5** | **78.2** | **4.3** | 170.1 | 11.1 | 1.5 |

‡ Inference speed (IS) is standardized to 1.0 = 1770 ms.

Table 2: Segmentation performance and generalizability on AMOS-CT and AMOS-MRI subsets for four training settings. (1) CT: single-modality CT training; (2) CT→MRI: CT pre-training followed by MRI fine-tuning; (3) MRI: single-modality MRI training; (4) CT+MRI: joint multi-modal training. Methods marked with (*) are initialized with pre-trained weights.

| Method | mDSC (%) ↑ | | | | mHD95 (mm) ↓ | | | |
|---|---|---|---|---|---|---|---|---|
| | CT | CT → MRI | MRI | CT + MRI | CT | CT → MRI | MRI | CT + MRI |
| Hermes [18] | 85.3 | 84.8 | 80.7 | 82.9 | **11.3** | 8.2 | 13.0 | 7.3 |
| Swin SMT [38] | 85.7 | 83.2 | 63.2 | 81.2 | 47.2 | 9.0 | 22.5 | 17.7 |
| VSmTrans [30] | 85.3 | 84.0 | 74.6 | 78.0 | 39.1 | 14.1 | 18.5 | 15.2 |
| uC 3DU-Net [22] | 82.7 | 84.5 | 68.6 | 84.1 | 19.1 | 9.1 | 16.9 | 12.2 |
| Swin UNETR [47] | 84.3 | 84.2 | 75.0 | 81.2 | 43.6 | 13.3 | 14.7 | 12.4 |
| SegMamba [54] | 86.0 | 84.4 | 80.2 | 84.7 | 26.3 | 8.2 | 12.6 | 7.5 |
| SuPreM†(*) [28] | 86.0 | 83.9 | 70.3 | 83.5 | 16.5 | 8.6 | 18.2 | 11.8 |
| **Mamba-HoME** | 86.3 | 84.8 | 81.0 | 85.1 | 18.7 | 8.1 | 11.7 | 7.4 |
| **Mamba-HoME** (*) | **87.3** | **85.0** | **82.3** | **86.4** | 12.2 | **8.0** | **11.0** | **7.2** |

† SuPreM was pre-trained on the AMOS-CT training set, while VoCo-B was pre-trained on both the training and validation sets. Here, we removed VoCo-B from the benchmark as it may lead to an unfair comparison.

## 3.1 Datasets

**Pre-training.** We utilize large-scale, publicly available datasets across two imaging modalities: AbdomenAtlas 1.1 [40, 27] for CT scans and TotalSegmentator MRI [1] for MRI scans. Both datasets provide high-quality, voxel-wise anatomical annotations of abdominal structures, ensuring a diverse and robust representation for multi-modal pre-training.

**Training and fine-tuning.** We use datasets covering three primary 3D medical imaging modalities, including publicly available PANORAMA (CT) [2], AMOS (CT and MRI) [23], FeTA 2022 (fetal MRI) [37], MVSeg (3D US) [10], and an in-house CT dataset. A detailed description of the datasets can be found in Appendix A.

## 3.2 Implementation details

The experiments were conducted on a workstation equipped with $8 \times$ NVIDIA H100 GPUs. For implementation, we employ Python 3.11, PyTorch 2.4 [35], and MONAI 1.3.0 within a Distributed Data-Parallel (DDP) training setup.

All training is performed using the $\mathcal{L}_{\text{DiceCE}}$ loss function and the AdamW optimizer, with an initial learning rate of 1e-4 controlled by a cosine annealing scheduler [32], a weight decay of 1e-5, and a batch size of 2. All models are trained with 32-bit floating-point precision to ensure numerical stability and to standardize the training process across all experiments. A detailed description of the implementation can be found in Appendix B.

## 3.3 Comparison with state-of-the-art methods

We compare our proposed Mamba-HoME with eight state-of-the-art approaches for 3D medical image segmentation, including uC 3DU-Net [22], Swin SMT [38], VoCo-B [53], SuPreM [28], Hermes [18], Swin UNETR [47], VSmTrans [30], and SegMamba (baseline) [54], across four publicly available and one in-house dataset, covering diverse anatomical structures and imaging modalities, such as CT, MRI, and 3D US. Notably, both VoCo-B and SuPreM are pre-trained on large-scale CT scans using self-supervised and supervised learning approaches, respectively. Additionally, we evaluate Mamba-HoME trained from scratch against Mamba-HoME pre-trained with a supervised learning approach to assess the impact of pre-training on segmentation performance. Detailed quantitative and more qualitative results for benchmarking datasets can be found in Appendix C, and Appendix E, respectively.

Table 3: Segmentation performance on FeTA via 5-fold cross-validation. Methods marked with (*) are initialized with pre-trained weights.

| Method | mDSC (%) ↑ | mHD95 (mm) ↓ |
|---|---|---|
| VoCo-B (*) [53] | 86.0 | 4.0 |
| Hermes [47] | 86.5 | 4.0 |
| Swin SMT [38] | 85.9 | 2.4 |
| VSmTrans [30] | 86.1 | 2.3 |
| uC 3DU-Net [22] | 85.9 | 3.5 |
| Swin UNETR [47] | 86.2 | 2.5 |
| SegMamba [54] | 85.9 | 3.5 |
| SuPreM (*) [28] | 85.3 | 3.6 |
| **Mamba-HoME** | 87.5 | 2.1 |
| **Mamba-HoME** (*) | **87.7** | **2.0** |

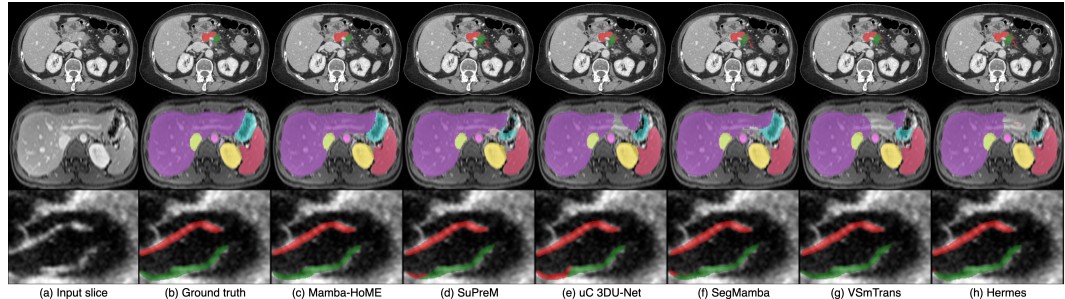

| (a) Input slice | (b) Ground truth | (c) Mamba-HoME | (d) SuPreM | (e) uC 3DU-Net | (f) SegMamba | (g) VSmTrans | (h) Hermes |

Figure 2: Qualitative segmentation results from top to bottom: CT, MRI, and 3D US. From left to right, each column shows the input slice, ground truth, the proposed Mamba-HoME, and the five next best-performing methods.

**Quantitative results.** Results for our proposed method, Mamba-HoME, on the PANORAMA and in-house datasets are shown in Table 1. Results for other modalities, including AMOS (CT/MRI), FeTA 2022 (fetal MRI), and MVSeg (3D US), are presented in Table 2, Table 3, and Table 4, respectively.

Our proposed method, Mamba-HoME, demonstrates consistent performance improvements over state-of-the-art baselines across all benchmark datasets and three imaging modalities. Evaluated under two distinct configurations (scratch and pretrained), Mamba-HoME achieves superior segmentation accuracy, obtaining the best results in terms of both DSC and HD95. Despite having a relatively large number of parameters (170.1M) compared to competing methods, it exhibits low GPU memory usage during inference (see Table 1), a crucial advantage for processing high-resolution 3D medical data. Although inference is approximately 30% slower than the baseline, the performance gains present a compelling trade-off between accuracy and efficiency. The Wilcoxon signed-rank test indicates a significant difference between Mamba-HoME and other state-of-the-art methods, with a significance threshold of $p < 0.05$.

Table 4: Segmentation performance on the MVSeg test set. Methods marked with (*) are initialized with pre-trained weights.

| Method | mDSC (%) ↑ | mHD95 (mm) ↓ |
|---|---|---|
| VoCo-B (*) [53] | 84.3 | 5.1 |
| Hermes [18] | 83.5 | 5.3 |
| Swin SMT [38] | 83.4 | 4.9 |
| VSmTrans [30] | 84.4 | 6.2 |
| uC 3DU-Net [22] | 83.9 | 4.7 |
| Swin UNETR [47] | 84.4 | 4.8 |
| SegMamba [54] | 83.8 | 5.8 |
| SuPreM (*) [28] | 84.3 | 5.0 |
| **Mamba-HoME** | 84.8 | 4.3 |
| **Mamba-HoME** (*) | **85.0** | **4.1** |

**Qualitative results.** Figure 2 presents a qualitative comparison of our proposed Mamba-HoME method against the five top-performing baselines across three primary 3D medical imaging modalities: CT, MRI, and US. These modalities exhibit different organ contrasts, noise levels, and resolutions. Mamba-HoME demonstrates consistent improvements in segmentation quality across these scenarios. In the first row, it effectively handles small and closely located structures, showing precise boundary delineation while reducing common artifacts seen in baseline predictions. The second row highlights its capability to accurately segment organs of various shapes and sizes, even under low image quality conditions, with reduced susceptibility to over- or under-segmentation. The third row illustrates Mamba-HoME's robustness in handling noisy and low-resolution data, maintaining clear and anatomically accurate boundaries.

### 3.4 Ablation studies

In this section, we investigate the impact of several factors on the performance of Mamba-HoME: (1) the parameters of the HoME layer, including the number of experts in the first ($E_1$) and second ($E_2$) levels, the group size ($K$), and the number of slots per expert ($S$); (2) the effect of Dynamic Tanh normalization compared to Layer Normalization, specifically its influence on training and validation speed in SSMs and overall performance; and (3) the impact of the pre-trained model in a supervised learning approach. For each configuration, we evaluate the number of model parameters, GPU memory usage, and average DSC across three datasets[2]. Further details regarding these ablation studies are provided in Appendix D.

---

[2]In these experiments, we use PANORAMA (PANO), AMOS-CT (AMOS), and FeTA.

**Effect of the number of experts.** We evaluate the impact of varying the number of experts at each encoder stage ($i$), where $i \in \{1, 2, 3, 4\}$, in a two-level HoME layer. For a fair comparison, we keep the group size constant ($K \in \{2048, 1024, 512, 256\}$) and set the number of slots to $S = 4$. Table 5 shows that the configuration with $E_1 \in \{4, 8, 12, 16\}$ experts at the first level and $E_2 \in \{8, 16, 24, 32\}$ experts at the second level achieves the best trade-off between segmentation performance and parameter efficiency. This setup also requires the fewest parameters and the lowest GPU memory usage.

Table 5: Quantitative segmentation performance of Mamba-HoME with varying numbers of experts at each encoder stage $i$.

| # | Number of Experts (E) | Params (M) ↓ | GPU (G) ↓ | mDSC (%) ↑ PANO | AMOS | FeTA |
|---|---|---|---|---|---|---|
| 1 | $E_1 = [4, 8, 12, 16]$ $E_2 = [8, 16, 24, 32]$ | **170.1** | **11.1** | **77.5** | **86.3** | **87.5** |
| 2 | $E_1 = [8, 16, 24, 48]$ $E_2 = [8, 16, 24, 48]$ | 277.8 | 12.2 | 76.3 | 86.2 | 87.2 |
| 3 | $E_1 = [8, 16, 24, 48]$ $E_2 = [16, 32, 48, 96]$ | 359.2 | 12.9 | 75.8 | 85.9 | 87.4 |
| 4 | $E_1 = [16, 32, 48, 64]$ $E_2 = [16, 32, 48, 64]$ | 367.0 | 13.0 | **77.5** | 86.1 | 87.4 |

**Effect of the group size.** We evaluate the impact of the group size at each encoder stage ($i$), where $i \in \{1, 2, 3, 4\}$, in a two-level HoME layer. For a fair comparison, we keep the number of experts constant ($E_1 \in \{4, 8, 12, 16\}$ and $E_2 \in \{8, 16, 24, 32\}$) and set the number of slots to $S = 4$. Table 6 shows that Mamba-HoME achieves optimal performance with group sizes $K \in \{2048, 1024, 512, 256\}$, while also minimizing GPU memory usage.

Table 6: Quantitative segmentation performance of Mamba-HoME with varying group sizes at each encoder stage $i$.

| # | Group size (K) | Params (M) ↓ | GPU (G) ↓ | mDSC (%) ↑ PANO | AMOS | FeTA |
|---|---|---|---|---|---|---|
| 1 | [1024, 512, 256, 128] | 170.1 | 11.1 | 77.2 | 86.1 | 87.4 |
| 2 | [2048, 1024, 512, 256] | 170.1 | 11.1 | **77.5** | **86.3** | 87.5 |
| 3 | [2048, 1024, 512, 256]† | 277.8 | 12.2 | 76.8 | 86.2 | 87.3 |
| 4 | [4096, 2048, 1024, 512] | 170.1 | 11.1 | 77.4 | 86.1 | 87.4 |

† The number of experts in the first and second levels of the HoME layer are equal ($E_1 = E_2 = [8, 16, 24, 48]$).

**Effect of the number of slots per expert.** We examine the impact of the number of slots ($S$) per expert ($E$) at each encoder stage ($i$), where $i \in \{1, 2, 3, 4\}$, in a two-level HoME layer. For a fair comparison, we keep the number of experts constant ($E_1 \in \{4, 8, 12, 16\}$ and $E_2 \in \{8, 16, 24, 32\}$) and the number of groups fixed ($K \in \{2048, 1024, 512, 256\}$). Table 7 shows that Mamba-HoME achieves optimal performance at $S = 4$ across all evaluated datasets, representing a sweet spot for the number of slots per expert. Variations in the number of slots ($S \in \{1, 2, 8\}$) do not yield significant performance improvements, with other slot counts resulting in suboptimal performance while keeping the number of parameters and GPU memory constant.

Table 7: Quantitative segmentation performance of Mamba-HoME with varying numbers of slots per expert.

| # | Slots (S) | mDSC (%) ↑ PANO | AMOS | FeTA |
|---|---|---|---|---|
| 1 | 1 | 76.2 | 85.9 | 87.3 |
| 2 | 2 | 77.2 | 86.0 | 87.4 |
| 3 | 4 | **77.5** | **86.3** | **87.5** |
| 4 | 8 | 76.5 | 86.1 | 87.4 |

**Effect of Dynamic Tanh normalization in SSMs.** While DyT accelerates CNN- and Transformer-based architectures [58], we investigate its effectiveness in SSM-based architectures compared to Layer Normalization [7]. As shown in Table 8, segmentation performance remains largely unchanged, but DyT improves both training and inference speed by approximately 6% based on experimental runtime measurements[3].

**Impact of supervised pre-training.** We evaluate the efficacy of the proposed Mamba-HoME model, pre-trained under a supervised learning paradigm on publicly available datasets, including 8,788 CT and 616 MRI scans with voxel-wise annotations. The evaluation spans three primary 3D medical imaging modalities and various anatomical regions. Overall, Mamba-HoME outperforms state-of-the-art methods, surpassing existing approaches in both DSC and HD95 across all evaluated datasets. These quantitative results highlight the effectiveness of supervised pre-training in enhancing segmentation accuracy and robustness. A key feature of Mamba-HoME is its cross-modal generalization,

Table 8: Quantitative segmentation performance of Mamba-HoME trained with Layer Normalization (LN) and Dynamic Tanh (DyT).

| # | Dataset | LN | DyT | mDSC (%) ↑ |
|---|---|---|---|---|
| 1 | PANO | ✓ | ✗ | 77.4 |
| 2 | | ✗ | ✓ | **77.5** |
| 3 | AMOS | ✓ | ✗ | 86.2 |
| 4 | | ✗ | ✓ | **86.3** |
| 5 | FeTA | ✓ | ✗ | **87.5** |
| 6 | | ✗ | ✓ | 87.4 |

---

[3]For each dataset in this experiment, we adopt the same settings described in Appendix B.

enabled by modality-agnostic feature representations. Pre-trained on CT and MRI scans, the model demonstrates superior adaptability to specialized tasks, such as fetal brain MRI segmentation in the FeTA dataset and 3D ultrasound mitral valve leaflet segmentation in the MVSeg dataset.

This adaptability highlights Mamba-HoME's ability to mitigate challenges posed by variations in modality, resolution, and clinical context. Moreover, its consistently high performance across heterogeneous datasets underscores its potential for practical deployment, where robust, modality-agnostic feature representations and precise segmentation are essential for scalable, real-world medical imaging applications. As shown in Figure 3, supervised pre-training significantly improves Mamba-HoME's performance compared to training from scratch or baseline methods, reducing artifacts and enhancing boundary segmentation for objects of varying sizes across the three primary 3D medical imaging modalities.

### 3.5 Generalizability analysis

To evaluate generalizability, we compare the proposed Mamba-HoME with several state-of-the-art networks. Specifically, we investigate four configurations on the AMOS dataset: (1) training solely on CT, (2) pre-training all models on CT and fine-tuning on MRI, (3) training solely on MRI, and (4) joint training on both CT and MRI. Table 2 shows that Mamba-HoME demonstrates superior generalizability across modalities compared to other models. Trained from scratch and further pre-trained on large-scale CT and MRI datasets, Mamba-HoME exhibits strong cross-modal generalizability to 3D ultrasound data, a modality with

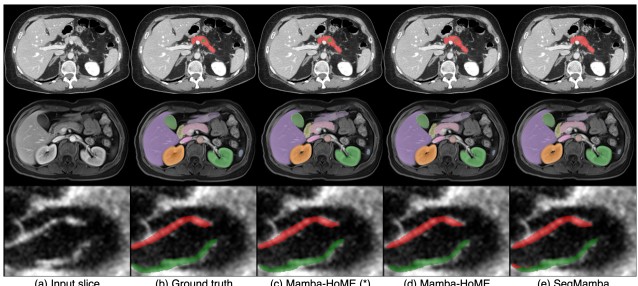

(a) Input slice  (b) Ground truth  (c) Mamba-HoME (*)  (d) Mamba-HoME  (e) SegMamba

Figure 3: Qualitative segmentation results from top to bottom: CT, MRI, and 3D US. From left to right, each column shows the input slice, ground truth, our proposed pre-trained Mamba-HoME, Mamba-HoME trained from scratch, and the baseline SegMamba.

distinct challenges such as high noise and lower resolution. Leveraging robust, modality-agnostic feature representations, the pre-trained model adapts to 3D ultrasound via efficient fine-tuning, outperforming state-of-the-art methods in both DSC and boundary HD95 metrics, as shown in Table 4. Qualitative results in Figure 2 further illustrate its ability to handle ultrasound-specific artifacts. This cross-modal transferability highlights the model's versatility across diverse imaging modalities. Moreover, Mamba-HoME demonstrates strong generalizability to external datasets within the same modality, especially MSD Pancreas and in-house CT dataset for PDAC and pancreas segmentation, outperforming several state-of-the-art methods in both DSC and HD95 metrics (see Table 11). Detailed results for the generalizability analysis can be found in Appendix F.

## 4 Conclusions

In this work, we introduce the Hierarchical Soft Mixture-of-Experts (HoME), a two-level token-routing MoE layer designed to efficiently capture local-to-global pattern hierarchies. We integrate HoME with Mamba in the Mamba-HoME architecture, enabling efficient long-sequence processing. Comprehensive experiments show that Mamba-HoME outperforms several state-of-the-art methods and generalizes well across the three primary 3D medical imaging modalities.

**Limitations.** Scalability to large-scale medical datasets (e.g., >10,000 scans) remains unexplored, limiting our understanding of Mamba-HoME's generalization across diverse image distributions. Although the model is pre-trained on a large multimodal dataset using supervised learning, its behavior under large-scale self-supervised learning (e.g., >200,000 scans) has not yet been studied. We identify this as a promising direction for future work to enhance Mamba-HoME's ability to capture complex patterns in unlabeled medical images. Key challenges include variations in image resolution, noise, contrast, field-of-view, acquisition techniques, and spatial-temporal dependencies.

## Acknowledgments

We acknowledge the use of the HPC cluster at Helmholtz Munich for the computational resources used in this study. Ewa Szczurek acknowledges the support from the Polish National Science Centre SONATA BIS grant no. 2020/38/E/NZ2/00305.

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

# Appendix

## Table of Contents

# A Datasets

A core objective of this work is to evaluate the robustness and generalizability of Mamba-HoME across a wide range of clinical scenarios. To this end, we conduct a comprehensive evaluation using datasets from three primary 3D medical imaging modalities: CT, MRI, and US. This multi-modal cohort provides a diverse spectrum of imaging characteristics, ranging from high-resolution structural details to low-resolution volumes with varying levels of noise and artifacts. Table 9 provides a detailed overview of the datasets utilized for pre-training, supervised training, fine-tuning, and testing.

Table 9: An overview of the datasets used for pre-training, training, fine-tuning, and testing. These datasets, spanning three modalities (CT, MRI, and US), cover diverse anatomical structures and lesions. Please note that all datasets configured in training mode were utilized for fine-tuning.

| No. | Dataset | Modality | Body part | Mode | Label type | Pre-training | Train | Test |
|---|---|---|---|---|---|---|---|---|
| 1. | AbdomenAtlas 1.1 [40, 27, 28] | CT | Abdomen | Pre-training | Voxel-wise | 8,788 (-474)[4] | - | - |
| 2. | TotalSegmentator MRI [1] | MRI | Whole-body | Pre-training | Voxel-wise | 616 | - | - |
| 3. | PANORAMA [2] | CT | Abdomen | Training | Voxel-wise | - | 1,964 | 334 |
| 4. | AMOS [23] | CT-MRI | Abdomen | Training | Voxel-wise | - | 240 | 120 |
| 5. | FeTA 2022 [37] | MRI | Fetal brain | Training | Voxel-wise | - | 120 | - |
| 6. | MVSeg [10] | US | Heart | Training | Voxel-wise | - | 110 | - |
| 7. | In-house CT | CT | Abdomen | Test | Voxel-wise | - | - | 60 |
| | | | | | **Total** | **9,404** | **2,434** | **514** |

## A.1 Pre-training

For pre-training, we utilize two manually voxel-wise annotated, publicly available large-scale datasets across two modalities: AbdomenAtlas 1.1 [27, 40, 28], comprising multi-phase CT scans, and TotalSegmentator MRI [1], consisting of diverse MRI volumes. The mapping of classes across both modalities used for model pre-training is detailed in Table 10.

**AbdomenAtlas.** This large-scale dataset comprises 9,262 CT scans featuring voxel-wise annotations for 25 anatomical structures. The collection encompasses a diverse range of acquisition phases, including non-contrast, arterial, portal-venous, and delayed phases. Each CT volume contains between 24 and 2,572 slices, with in-plane resolutions ranging from $188 \times 79$ to $971 \times 651$ pixels. The voxel spatial resolution ranges from $([0.38 \sim 1.5] \times [0.38 \sim 3.0] \times [0.3 \sim 8.0])$ mm$^3$, with a mean resolution of $0.84 \times 0.84 \times 2.4$ mm$^3$. To ensure the integrity of our evaluation and provide a fair comparison, we excluded cases that overlap with our training and test sets. Specifically, we removed 194 cases from MSD Pancreas, 43 from NIH Pancreas, and 200 training cases from the AMOS-CT dataset, resulting in a curated pre-training cohort.

**TotalSegmentator MRI.** This dataset consists of 616 MRI volumes featuring 50 anatomical structures manually annotated at the voxel level. For the pre-training phase, we specifically select a subset of 22 classes that align with the labels provided in the AbdomenAtlas dataset. Each MRI scan contains between 5 and 1,915 slices, with a highly variable voxel spatial resolution ranging from $([0.17 \sim 20.0] \times [0.17 \sim 25.0] \times [0.17 \sim 28.0])$ mm$^3$ and a mean resolution of $1.28 \times 1.86 \times 2.81$ mm$^3$.

## A.2 Training, fine-tuning, and test

To validate the efficiency of the Mamba-HoME, we conduct comprehensive experiments on both publicly available and in-house datasets across three primary modalities. For CT, we use PANORAMA [2], AMOS-CT [23], and a private dataset for segmentation and diagnosis of pancreatic ductal adenocarcinoma (PDAC) in abdominal CT. For MRI, we use AMOS-MRI [23], and FeTA 2022, which includes fetal brain MRI [36, 37]. For US, we use MVSeg [10].

**PANORAMA.** This dataset consists of 2,238 multi-center contrast-enhanced computed tomography (CECT) scans acquired in the portal-venous phase. It includes 1,964 newly acquired scans from five European centers (located in the Netherlands, Norway, and Sweden), as well as publicly available data from two medical centers in the United States: the NIH [43] and MSD Pancreas [3]. The cohort

---

[4]Please note that, for a fair comparison, we excluded 200 cases from AMOS, 194 from MSD Pancreas, and 80 from NIH, respectively. These scans correspond to the original AMOS dataset and partly to the PANORAMA dataset.

Table 10: Class mapping for anatomical structures in CT scans (AbdomenAtlas) and MRI scans (TotalSegmentator MRI).

| Index | Class | Index | Class |
|---|---|---|---|
| 0 | Background | 13 | Celiac trunk |
| 1 | Aorta | 14 | Colon |
| 2 | Gall bladder | 15 | Duodenum |
| 3 | Kidney (left) | 16 | Esophagus |
| 4 | Kidney (right) | 17 | Femur (left) |
| 5 | Liver | 18 | Femur (right) |
| 6 | Pancreas | 19 | Hepatic vessel |
| 7 | Postcava | 20 | Intestine |
| 8 | Spleen | 21 | Lung (left) |
| 9 | Stomach | 22 | Lung (right) |
| 10 | Adrenal gland (left) | 23 | Portal vein & splenic vein |
| 11 | Adrenal gland (right) | 24 | Prostate |
| 12 | Bladder | 25 | Rectum |

comprises 676 PDAC (pancreatic ductal adenocarcinoma) and 1,562 non-PDAC cases. While the original dataset contains six voxel-wise labels, including PDAC lesion, veins, arteries, pancreatic parenchyma, pancreatic duct, and common bile duct, we specifically utilize the PDAC lesion label and merge the pancreatic parenchyma and pancreatic duct classes into a single *pancreas* label. Each CT volume contains between 37 and 1,572 slices, with an in-plane resolution ranging from $512 \times 512$ to $1024 \times 1024$ pixels. The voxel spatial resolution ranges from ($[0.31 \sim 1.03] \times [0.31 \sim 1.03] \times [0.45 \sim 5.0]$) mm$^3$, with a mean resolution of $0.75 \times 0.75 \times 2.0$ mm$^3$.

**AMOS.** This dataset comprises a total of 600 multi-modal scans (500 CT and 100 MRI). Ground truth annotations are provided for the training and validation cohorts, which include 240 and 120 scans, respectively. Each scan features 15 anatomical structures, manually annotated at the voxel level. For our experiments on **AMOS-CT**, we utilize 200 scans for training and 100 for testing. For **AMOS-MRI**, we employ 40 scans for training and 20 for testing. The CT volumes consist of $68 \sim 353$ slices, with a voxel spatial resolution ranging from ($[0.45 \sim 1.07] \times [0.45 \sim 1.07] \times [1.25 \sim 5.0]$) mm$^3$ and a mean of $0.70 \times 0.70 \times 4.20$ mm$^3$. The MRI volumes contain $60 \sim 168$ slices, with a voxel resolution ranging from ($[0.70 \sim 1.95] \times [0.70 \sim 1.95] \times [1.09 \sim 3.0]$) mm$^3$ and a mean of $1.10 \times 1.10 \times 2.46$ mm$^3$.

**FeTA 2022.** This dataset consists of 120 fetal MRI scans for brain tissue segmentation, collected from two prominent medical institutions: the University Children's Hospital Zurich and the Medical University of Vienna. The cohort includes 80 cases from Zurich and 40 from Vienna. It features seven manually annotated tissue classes provided at the voxel level: external cerebrospinal fluid, gray matter, white matter, ventricles, cerebellum, deep gray matter, and the brainstem. Each fetal MRI volume contains 256 slices, with an isotropic voxel spatial resolution ranging from ($[0.43 \sim 1.0] \times [0.43 \sim 1.0] \times [0.43 \sim 1.0]$) mm$^3$ and a mean resolution of $0.67 \times 0.67 \times 0.67$ mm$^3$.

**MVSeg.** This dataset consists of 175 transesophageal echocardiography (TEE) 3D ultrasound scans dedicated to mitral valve segmentation. Voxel-wise ground truth annotations are provided for both the posterior and anterior leaflets. The data were acquired at King's College Hospital, London, UK. Each 3D ultrasound volume contains 208 slices, with a voxel spatial resolution ranging from ($[0.20 \sim 0.63] \times [0.31 \sim 0.90] \times [0.13 \sim 0.39]$) mm$^3$ and a mean resolution of $0.38 \times 0.56 \times 0.23$ mm$^3$.

**In-house CT.** This dataset consists of 60 CECT scans of histopathology-confirmed PDAC cases, acquired from 40 different medical centers. Each volume contains $68 \sim 875$ slices with an in-plane resolution of $512 \times 512$ pixels. The voxel spatial resolution ranges from ($[0.47 \sim 0.98] \times [0.47 \sim 0.98] \times [0.5 \sim 5.0]$) mm$^3$. These scans include voxel-wise annotations for both the pancreas and the PDAC lesion. The diameter of the PDAC lesions ranges from $1.25 \sim 6.06$ cm, with a mean of $3.45 \pm 1.20$ cm. To ensure high-quality ground truth, the data were initially annotated by two junior radiologists using 3D Slicer and subsequently reviewed and refined by two domain experts, each with over 30 years of clinical experience.

# B  Experimental setup

## B.1  Pre-training

**AbdomenAtlas and TotalSegmentator MRI.** For both the AbdomenAtlas 1.1 and TotalSegmentator MRI datasets, all volumes are resampled to an anisotropic resolution of $0.8 \times 0.8 \times 3.0$ mm$^3$. Prior to training, the data are partitioned into training and validation sets using a stratified 85:15 split to maintain a balanced distribution of CT and MRI modalities. Intensity preprocessing for CT scans involves clipping Hounsfield Units (HU) to the range of $[-175, 250]$, while MRI intensities are clipped to $[0, 1000]$; both modalities are subsequently min-max scaled to the interval $[0, 1]$. Mamba-HoME is trained with a patch size of $96 \times 96 \times 96$ for 800 epochs. To enhance model robustness, we employ on-the-fly stochastic augmentations, including random scaling, rotation, flipping, brightness and contrast adjustment, Gaussian smoothing, and additive Gaussian noise. The pre-training phase was completed in approximately 7 days using $8\times$ NVIDIA H100 80GB GPUs.

## B.2  Training and fine-tuning

**PANORAMA.** All CT volumes are resampled to an anisotropic resolution of $0.8 \times 0.8 \times 3.0$ mm$^3$. Voxel intensities are clipped to a Hounsfield Unit (HU) range of $[-175, 250]$ and subsequently min-max scaled to the interval $[0, 1]$. We employ a stratified 80:20 split, resulting in 1,571 training and 393 validation scans, thereby preserving the class distribution of PDAC versus non-PDAC cases. The model is fine-tuned for 500 epochs using a consistent training patch size of $192 \times 192 \times 48$ voxels. To maximize robustness, we utilize on-the-fly stochastic augmentations, including random scaling, rotation, flipping, brightness/contrast adjustments, and the application of Gaussian smoothing and noise. During the inference stage, a sliding-window strategy is employed with a corresponding crop size of $192 \times 192 \times 48$ and an overlap ratio of $0.5$, incorporating a Gaussian importance weighting filter to ensure seamless patch aggregation and minimize boundary artifacts.

**AMOS.** For both modalities, all volumes are resampled to an isotropic resolution of $1.5 \times 1.5 \times 1.5$ mm$^3$. Intensity preprocessing follows the protocol described in the pre-training phase: CT scans are clipped to a Hounsfield Unit (HU) range of $[-175, 250]$, while MRI scans are clipped to $[0, 1000]$. Both modalities are subsequently min-max scaled to the interval $[0, 1]$. The original training set is partitioned into training and validation subsets using a randomized 80:20 split. Across the training, fine-tuning, and inference stages, we maintain a consistent spatial context by utilizing a patch size of $128 \times 128 \times 128$. For both validation and final testing, a sliding-window inference strategy is employed with a default overlap ratio of $0.5$, incorporating a Gaussian weighting filter to ensure smooth patch aggregation. Data robustness is further enhanced via on-the-fly augmentations, including random scaling, rotation, flipping, brightness/contrast adjustments, and the application of Gaussian smoothing and additive noise.

**FeTA 2022.** All MRI volumes are resampled to an isotropic resolution of $0.8 \times 0.8 \times 0.8$ mm$^3$. Intensities are clipped to the range of $[0, 1000]$ and subsequently min-max scaled to the interval $[0, 1]$. We evaluate model performance using a 5-fold cross-validation strategy, with each fold trained for 300 epochs. Throughout the training and inference phases, we utilize a consistent patch size of $128 \times 128 \times 128$ voxels. To maximize generalizability in the presence of fetal motion and anatomical variability, we employ comprehensive on-the-fly stochastic augmentations, including random scaling, rotation, flipping, brightness/contrast adjustments, Gaussian smoothing, additive noise, and affine transformations.

**MVSeg.** All 3D ultrasound volumes are resampled to an isotropic resolution of $0.5 \times 0.5 \times 0.5$ mm$^3$. Intensities are clipped to the range of $[0, 255]$ and subsequently min-max scaled to the interval $[0, 1]$. We partition the dataset into independent training, validation, and testing sets consisting of 105, 30, and 40 scans, respectively. The model is trained for 500 epochs using a consistent patch size of $128 \times 128 \times 128$ for both training and inference. To address the inherent speckle noise and artifacts typical of ultrasound imaging, we employ extensive on-the-fly stochastic augmentations, including random scaling, rotation, flipping, brightness/contrast adjustments, Gaussian smoothing, additive noise, and affine transformations.

## B.3   Evaluation metrics

**Dice Similarity Coefficient (DSC).** The DSC is a commonly used metric to evaluate the segmentation performance of a model in multi-class settings, especially in medical imaging. It measures the overlap between predicted and ground truth segmentations, providing an aggregate assessment across multiple classes. Given a segmentation task with $C$ classes, let $\mathbf{p}_i \in \mathbb{R}^C$ and $\mathbf{g}_i \in \mathbb{R}^C$ be the one-hot encoded predicted and ground truth vectors at voxel $i$, respectively. The DSC for class $c$ is computed as:

$$DSC_c = \frac{2\sum_i p_{i,c} g_{i,c}}{\sum_i p_{i,c} + \sum_i g_{i,c}}, \tag{18}$$

where $p_{i,c}$ and $g_{i,c}$ represent the predicted and ground truth binary masks for class $c$ at voxel $i$, respectively. The mean DSC (mDSC) across all $C$ classes is computed as follows:

$$mDSC = \frac{1}{C}\sum_{c=1}^{C} DSC_c. \tag{19}$$

This metric ensures that each class contributes equally to the final score, regardless of class imbalance.

**95th percentile Hausdorff Distance (HD95).** The 95th percentile Hausdorff Distance (HD95) is a robust evaluation metric used to quantify the spatial similarity between predicted and ground truth segmentations. Unlike the standard Hausdorff Distance, which is highly sensitive to outliers by measuring the maximum distance, the HD95 considers the $95^{th}$ percentile of distances to provide a more stable measure of boundary agreement.

For a given class $c$, let $S_c^P$ and $S_c^G$ represent the sets of surface boundary points for the predicted and ground truth masks, respectively. These surface points are extracted by identifying voxels that change during binary erosion.

1. **Directed Distance Sets:** We first compute the set of minimum Euclidean distances from every point in one set to the nearest point in the other:

$$D(S_c^P, S_c^G) = \{\min_{g \in S_c^G} \|p - g\|_2 \mid p \in S_c^P\} \tag{20}$$

$$D(S_c^G, S_c^P) = \{\min_{p \in S_c^P} \|g - p\|_2 \mid g \in S_c^G\} \tag{21}$$

2. **95th Percentile Calculation:** The bi-directional HD95 for class $c$ is defined as the maximum of the 95th percentiles of these two distance distributions:

$$HD95_c = \max\left(\text{percentile}_{95}(D(S_c^P, S_c^G)), \text{percentile}_{95}(D(S_c^G, S_c^P))\right) \tag{22}$$

3. **Mean HD95 (mHD95):** To evaluate the overall performance across all $C$ classes, the mean HD95 is calculated as the average of class-wise means:

$$mHD95 = \frac{1}{C}\sum_{c=1}^{C}\left(\frac{1}{N_c}\sum_{i=1}^{N_c} HD95_{c,i}\right), \tag{23}$$

where $N_c$ is the number of cases where class $c$ was present.

**Sensitivity and Specificity.** The sensitivity and specificity are crucial metrics for evaluating the performance of a model in detecting the presence of a condition at the patient level. In this setting, a segmentation model processes medical scans and outputs a binary classification for each patient: either *positive* (presence of the condition) or *negative* (absence of the condition).

Given a dataset of patients, let $TP$, $FP$, $FN$, and $TN$ denote the number of true positives, false positives, false negatives, and true negatives, respectively. The sensitivity (also known as recall or true positive rate) is defined as:

$$\text{Sensitivity} = \frac{TP}{TP + FN}, \tag{24}$$

where sensitivity measures the proportion of correctly identified positive patients out of all actual positive patients. The specificity (true negative rate) is given by:

$$\text{Specificity} = \frac{TN}{TN + FP}, \tag{25}$$

where specificity quantifies the proportion of correctly identified negative patients out of all actual negative patients.

These metrics provide a comprehensive assessment of the model's ability to detect the condition while avoiding false alarms, which is critical for clinical decision-making.

**Number of parameters.** The total number of trainable parameters in a neural network can be computed by summing the parameters across all layers and levels, as follows:

$$\text{Parameters} = \|\Theta\|_0 + \sum_{l=1}^{L} \sum_{k=1}^{L-l} \sum_{i=1}^{C_l} \sum_{j=1}^{C_k} A_{l,k}^{i,j} \|\theta_{l,k}^{i,j}\|_0, \tag{26}$$

where $\Theta$ represents the set of parameters from the backbone, $L$ is the total number of levels or layers in the network, $C_l$ and $C_k$ represent the number of input and output channels at each level $l$ and $k$, respectively, $A_{l,k}^{i,j}$ is a binary matrix indicating the presence of a weight connection between input channel $i$ at level $l$ and output channel $j$ at level $k$, $\|\theta_{l,k}^{i,j}\|_0$ represents the count of non-zero weights in the connection between input channel $i$ and output channel $j$.

This formulation takes into account the layer-wise parameters while incorporating the structured sparsity of weights within the network.

**Inference speed.** The inference speed measures the temporal latency required for the model to process a volumetric input and generate a segmented output. Given the use of a sliding window approach for high-resolution medical volumes, the average inference speed $\bar{T}$ is calculated across $N$ samples following a GPU warm-up phase to ensure hardware state stabilization. This metric is defined as:

$$\bar{T} = \frac{1}{N} \sum_{n=1}^{N} (t_{\text{end},n} - t_{\text{start},n}) \times 10^3, \tag{27}$$

where $t_{\text{start},n}$ and $t_{\text{end},n}$ represent the timestamps in seconds recorded immediately before the input tensor is moved to the computation device and immediately after the sliding window aggregator completes the reconstruction of the prediction volume, respectively. The constant $10^3$ scales the result to milliseconds (ms).

## C Additional experimental results

Table 11: Quantitative segmentation and detection results on the PANORAMA test sets. Performance is evaluated across the NIH (healthy controls only), MSD Pancreas, and an in-house dataset. We report mDSC (%) and mHD95 (mm) for the pancreas and pancreatic ductal adenocarcinoma (PDAC), alongside patient-level detection performance via Sensitivity (%) and Specificity (%). The best and second-best results are **bolded** and underlined, respectively. Methods marked with (*) are initialized with pre-trained weights. Note that SuPreM utilized a subset of the test data (MSD Pancreas and NIH) during its pre-training phase, which may impact the fairness of the comparison.

| Method | mDSC (%) ↑ | | | | | | | mHD95 (mm) ↓ | | PDAC Detection (%) ↑ | |
|---|---|---|---|---|---|---|---|---|---|---|---|
| | NIH | MSD Pancreas | | In-house | | Overall | | Overall | | Overall | |
| | Pancreas | PDAC | Pancreas | PDAC | Pancreas | PDAC | Pancreas | PDAC | Pancreas | Sensitivity | Specificity |
| VoCo-B (*) | 90.1 | 38.7 | 86.5 | 43.3 | 80.3 | 40.5 | 86.3 | 32.8 | 7.4 | 86.1 | 91.6 |
| Hermes | 92.0 | 40.0 | 86.5 | 58.3 | 80.8 | 48.2 | 87.8 | 21.4 | 5.1 | 87.3 | 94.0 |
| Swin SMT | 91.9 | 44.0 | 87.2 | 58.6 | 79.8 | 49.4 | 87.0 | 19.8 | 5.8 | 88.6 | 92.8 |
| VSmTrans | 92.0 | 47.1 | 87.3 | 56.1 | 80.5 | 50.3 | 87.2 | 18.2 | 5.2 | 89.9 | 90.4 |
| SegMamba | 92.7 | 47.9 | 88.8 | 54.0 | 81.9 | 49.7 | **88.5** | 19.5 | 4.3 | 84.8 | **95.2** |
| uC 3DU-Net | 92.6 | 49.0 | **88.9** | 56.8 | 80.2 | 52.0 | 88.2 | 16.9 | 4.5 | 80.7 | 89.2 |
| Swin UNETR | 91.9 | 38.8 | 87.4 | 58.7 | 81.3 | 46.3 | 87.4 | 24.1 | 6.2 | 87.3 | 90.4 |
| SuPreM (*) | 92.3 | 46.0 | 88.6 | 61.4 | 82.0 | 51.7 | 88.3 | 9.0 | 2.4 | 87.3 | 89.2 |
| **Mamba-HoME** | **92.9** | 51.2 | 88.4 | 60.8 | 81.7 | 54.8 | 88.3 | 8.4 | 3.2 | 90.4 | 92.8 |
| **Mamba-HoME (*)** | 92.6 | **53.6** | 88.5 | **61.7** | **82.3** | 56.7 | 88.5 | 6.2 | 1.9 | **92.8** | **95.2** |

Table 12: Quantitative segmentation results using 5-fold cross-validation on the FeTA 2022 dataset. Performance is reported as DSC (%) for each tissue class and summarized via mDSC (%) and mHD95 (mm) for the aggregate average across all folds. Tissue classes include: external cerebrospinal fluid (eCSF), gray matter (GM), white matter (WM), ventricles (V), cerebellum (C), deep gray matter (dGM), and brainstem (B). The best and second-best results are **bolded** and underlined, respectively. Methods designated with (*) are initialized with pre-trained weights.

| Method | DSC (%) ↑ | | | | | | | mDSC (%) ↑ | mHD95 (mm) ↓ |
|---|---|---|---|---|---|---|---|---|---|
| | eCSF | GM | WM | V | C | dGM | B | | |
| VoCo-B (*) | 78.9 | 73.4 | 90.2 | 87.5 | 87.0 | 87.6 | 83.8 | 86.0 | 4.0 |
| Swin SMT | 78.3 | 72.6 | 90.0 | 86.2 | 86.6 | 87.9 | 83.8 | 85.6 | 2.4 |
| Hermes | 79.3 | 74.2 | 90.7 | 87.6 | 87.1 | 88.8 | 84.7 | 86.5 | 4.0 |
| SegMamba | 79.6 | 73.7 | 90.6 | 87.1 | 87.5 | 88.8 | 85.2 | 86.5 | 5.8 |
| uC 3DU-Net | 79.1 | 73.2 | 90.3 | 87.4 | 86.0 | 88.0 | 83.4 | 85.9 | 3.5 |
| SuPreM (*) | 78.0 | 71.7 | 89.8 | 86.8 | 85.9 | 87.1 | 83.5 | 85.3 | 3.6 |
| VSmTrans | 79.0 | 73.1 | 90.6 | 87.2 | 87.2 | 88.3 | 84.2 | 86.1 | 2.3 |
| **Mamba-HoME** | 80.4 | 75.8 | 91.4 | 88.4 | 88.8 | **89.4** | **86.2** | 87.5 | 2.1 |
| **Mamba-HoME (*)** | **80.6** | **76.1** | **91.9** | **88.8** | **90.0** | 88.8 | 86.1 | **87.7** | **2.0** |

Table 13: Quantitative segmentation results on the MVSeg test set. Performance is reported as DSC (%) for the posterior leaflet (PL) and anterior leaflet (AL), with mDSC (%) and mHD95 (mm) representing the aggregate average across both classes. The best and second-best results are **bolded** and underlined, respectively. Methods marked with (*) are initialized with pre-trained weights.

| Method | DSC (%) ↑ | | mDSC (%) ↑ | mHD95 (mm) ↓ |
|---|---|---|---|---|
| | PL | AL | | |
| Swin SMT | 82.2 | 84.5 | 83.4 | 4.9 |
| Hermes | 82.9 | 84.1 | 83.5 | 5.3 |
| uC 3DU-Net | 82.9 | 85.0 | 83.9 | 4.7 |
| SuPreM (*) | 83.2 | 85.4 | 84.3 | 5.0 |
| VoCo-B (*) | 83.3 | 85.3 | 84.3 | 5.1 |
| Swin UNETR | 83.8 | 85.1 | 84.4 | 4.8 |
| SegMamba | 83.0 | 84.7 | 83.8 | 5.8 |
| VSmTrans | 83.4 | 85.4 | 84.4 | 6.2 |
| **Mamba-HoME** | 84.0 | 85.7 | 84.8 | 4.3 |
| **Mamba-HoME (*)** | **84.2** | **86.0** | **85.0** | **4.1** |

# D  Ablation studies

We present detailed quantitative results from our ablation studies, analyzing three key factors: (1) the number of experts per stage; (2) the group size; and (3) the number of slots assigned to each expert.

## D.1  Effect of the number of experts

For a fair comparison, we keep the group size constant ($K \in \{2048, 1024, 512, 256\}$) and set the number of slots to $S = 4$.

Table 14: Quantitative ablation study on the influence of the number of experts across the PANORAMA test sets. Performance is reported as DSC (%) for each dataset and summarized via mDSC (%) and mHD95 (mm) for the aggregate average across all cohorts (NIH, MSD, and in-house). The experts $E$ denotes the number of experts across the four encoder stages. The best and second-best results are **bolded** and underlined, respectively. Configurations marked with ($*$) indicate a doubled number of experts in the second hierarchical level.

| Experts (E) | DSC (%) ↑ | | | | | | | mDSC (%) ↑ | mHD95 (mm) ↓ |
| | NIH | MSD | | In-house | | Overall | | | |
| | Pancreas | PDAC | Pancreas | PDAC | Pancreas | PDAC | Pancreas | | |
|---|---|---|---|---|---|---|---|---|---|
| [4, 8, 12, 16]* | **92.9** | **51.2** | 88.4 | **60.8** | 81.7 | **54.8** | 88.3 | **77.5** | **4.8** |
| [8, 16, 24, 48] | 92.6 | 49.4 | 88.2 | 59.6 | 80.6 | 53.3 | 87.9 | 76.3 | 5.4 |
| [8, 16, 24, 48]* | 92.1 | 47.9 | 88.0 | 57.4 | 80.5 | 52.2 | 87.5 | 75.8 | 5.9 |
| [16, 32, 48, 64] | 92.7 | 51.0 | **88.5** | 60.4 | **81.9** | 54.5 | **88.4** | **77.5** | 4.9 |

Table 15: Quantitative ablation study on the influence of hierarchical expert configurations on the AMOS-CT validation set. Performance is reported as DSC (%) for each of the 15 abdominal organs, with mDSC (%) and mHD95 (mm) representing the aggregate average. Organs include: Spleen (Sp), Right Kidney (RK), Left Kidney (LK), Gallbladder (GB), Esophagus (Es), Liver (Li), Stomach (St), Aorta (Ao), Inferior Vena Cava (IVC), Pancreas (Pa), Right Adrenal Gland (RAG), Left Adrenal Gland (LAG), Duodenum (Du), Bladder (Bl), and Prostate/Uterus (Pr/Ut). The experts $E$ denotes the number of experts across the four encoder stages. The best and second-best results are **bolded** and underlined, respectively. Configurations marked with ($*$) indicate a doubled number of experts in the second hierarchical level.

| Experts (E) | DSC (%) ↑ | | | | | | | | | | | | | | | mDSC (%) ↑ | mHD95 (mm) ↓ |
| | Sp | RK | LK | GB | Es | Li | St | Ao | IVC | Pa | RAG | LAG | Du | Bl | Pr/Ut | | |
|---|---|---|---|---|---|---|---|---|---|---|---|---|---|---|---|---|---|
| [4, 8, 12, 16]* | **96.0** | 95.0 | 94.4 | 81.7 | 82.0 | 94.7 | **90.2** | **93.8** | **88.9** | 84.0 | 74.4 | 74.1 | **78.4** | 83.8 | 83.2 | **86.3** | **18.7** |
| [8, 16, 24, 48] | 95.7 | 93.7 | 94.0 | 81.0 | 81.5 | 95.3 | 89.3 | 93.2 | 88.4 | 83.4 | 73.8 | **75.0** | 77.6 | **87.6** | **83.3** | 86.2 | 19.6 |
| [8, 16, 24, 48]* | 94.7 | **95.4** | **95.0** | **81.6** | 81.4 | 95.4 | 87.0 | 93.4 | 87.8 | 83.4 | 73.8 | 74.4 | 77.0 | 86.0 | 82.2 | 85.9 | 20.2 |
| [16, 32, 48, 64] | 95.8 | 93.8 | 94.8 | 80.0 | **83.0** | **95.7** | 87.5 | 93.6 | 88.4 | **83.8** | **75.0** | 74.5 | 77.5 | 85.7 | 82.4 | 86.1 | 19.7 |

Table 16: Quantitative ablation study on the effect of the number of experts in the HoME layer across 5-fold cross-validation on the FeTA 2022 dataset. Performance is reported as DSC (%) for each fetal brain tissue class and summarized via mDSC (%) and mHD95 (mm) for the aggregate average across all folds. Tissue classes include: external cerebrospinal fluid (eCSF), gray matter (GM), white matter (WM), ventricles (V), cerebellum (C), deep gray matter (dGM), and brainstem (B). The experts $E$ denotes the number of experts across the four encoder stages. The best and second-best results are **bolded** and underlined, respectively. Configurations marked with ($*$) indicate a doubled number of experts in the second hierarchical level.

| Experts (E) | DSC (%) ↑ | | | | | | | mDSC (%) ↑ | mHD95 (mm) ↓ |
| | eCSF | GM | WM | V | C | dGM | B | | |
|---|---|---|---|---|---|---|---|---|---|
| [4, 8, 12, 16]* | 80.4 | **75.8** | 91.4 | 88.4 | **88.8** | **89.4** | 86.2 | **87.5** | **2.1** |
| [8, 16, 24, 48] | 80.4 | 75.1 | 91.1 | 88.3 | 88.5 | 88.6 | 86.6 | 87.2 | 2.4 |
| [8, 16, 24, 48]* | **80.7** | 75.5 | 91.1 | 88.4 | 88.5 | **89.4** | **86.9** | 87.4 | 2.3 |
| [16, 32, 48, 64] | 80.5 | 75.4 | **91.5** | **88.5** | 88.0 | 88.9 | 86.5 | 87.4 | 2.3 |

## D.2 Effect of the group size

For a fair comparison, we keep the number of experts constant ($E_1 \in \{4, 8, 12, 16\}$ and $E_2 \in \{8, 16, 24, 32\}$) and set the number of slots to $S = 4$.

Table 17: Quantitative ablation study on the effect of the group size ($K$) across the PANORAMA test sets. Performance is reported as DSC (%) for individual datasets and summarized via mDSC (%) and mHD95 (mm) for the aggregate average across all cohorts. The group size $K$ denotes the number of tokens per routing group across the four encoder stages. The best and second-best results are **bolded** and underlined, respectively. Configurations marked with (*) indicate a doubled number of experts in the second hierarchical level.

| Group size (K) | DSC (%) ↑ | | | | | | | mDSC (%) ↑ | mHD95 (mm) ↓ |
|---|---|---|---|---|---|---|---|---|---|
| | NIH | MSD | | In-house | | Overall | | | |
| | Pancreas | PDAC | Pancreas | PDAC | Pancreas | PDAC | Pancreas | | |
| [1024, 512, 256, 128] | 92.5 | 50.4 | 88.2 | 60.5 | 81.5 | 54.2 | 88.1 | 77.2 | 5.9 |
| [2048, 1024, 512, 256] | **92.9** | 51.2 | **88.4** | **60.8** | 81.7 | 54.8 | **88.3** | **77.5** | **4.8** |
| [2048, 1024, 512, 256]* | 92.7 | 50.3 | 88.3 | 56.3 | **81.7** | 52.6 | 88.2 | 76.8 | 6.3 |
| [4096, 2048, 1024, 512] | 92.5 | **51.9** | 88.1 | 60.6 | 81.5 | **55.2** | 88.0 | 77.4 | 5.0 |

Table 18: Quantitative ablation study on the effect of the group size ($K$) on the AMOS-CT validation set. Performance is reported as DSC (%) for each of the 15 abdominal organs, with mDSC (%) and mHD95 (mm) representing the aggregate average. Organs include: Spleen (Sp), Right Kidney (RK), Left Kidney (LK), Gallbladder (GB), Esophagus (Es), Liver (Li), Stomach (St), Aorta (Ao), Inferior Vena Cava (IVC), Pancreas (Pa), Right Adrenal Gland (RAG), Left Adrenal Gland (LAG), Duodenum (Du), Bladder (Bl), and Prostate/Uterus (Pr/Ut). The group size $K$ denotes the number of tokens per routing group across the four encoder stages. The best and second-best results are **bolded** and underlined, respectively. Configurations marked with (*) indicate a doubled number of experts in the second hierarchical level.

| Group size (K) | DSC (%) ↑ | | | | | | | | | | | | | | | mDSC (%) ↑ | mHD95 (mm) ↓ |
|---|---|---|---|---|---|---|---|---|---|---|---|---|---|---|---|---|---|
| | Sp | RK | LK | GB | Es | Li | St | Ao | IVC | Pa | RAG | LAG | Du | Bl | Pr/Ut | | |
| [1024, 512, 256, 128] | 95.5 | **95.4** | 94.9 | 81.5 | **82.1** | 89.1 | 86.4 | 93.4 | 87.6 | 82.8 | 73.1 | **75.1** | 76.1 | 87.5 | 82.1 | 86.1 | 19.2 |
| [2048, 1024, 512, 256] | **96.0** | 95.0 | 94.4 | **81.7** | 82.0 | 94.7 | **90.2** | **93.8** | **88.9** | **84.0** | 74.4 | 74.1 | **78.4** | 83.8 | **83.2** | **86.3** | **18.7** |
| [2048, 1024, 512, 256]* | 95.5 | 95.2 | **95.6** | 81.0 | **82.1** | 89.1 | 86.2 | 93.2 | 87.5 | 83.9 | 75.2 | **75.1** | 77.3 | 85.6 | 82.1 | 86.2 | 18.9 |
| [4096, 2048, 1024, 512] | 95.7 | 95.1 | 95.5 | 81.2 | 82.0 | 89.0 | 86.5 | 93.5 | 87.8 | 82.8 | 73.3 | 75.0 | 76.4 | **87.8** | 82.0 | 86.1 | 19.1 |

Table 19: Quantitative ablation study on the effect of the group size ($K$) across 5-fold cross-validation on the FeTA 2022 dataset. Performance is reported as DSC (%) for each fetal brain tissue class and summarized via mDSC (%) and mHD95 (mm) for the aggregate average across all folds. Tissue classes include: external cerebrospinal fluid (eCSF), gray matter (GM), white matter (WM), ventricles (V), cerebellum (C), deep gray matter (dGM), and brainstem (B). The group size $K$ denotes the number of tokens per routing group across the four encoder stages. The best and second-best results are **bolded** and underlined, respectively. Configurations marked with (*) indicate a doubled number of experts in the second hierarchical level.

| Group size (K) | DSC (%) ↑ | | | | | | | mDSC (%) ↑ | mHD95 (mm) ↓ |
|---|---|---|---|---|---|---|---|---|---|
| | eCSF | GM | WM | V | C | dGM | B | | |
| [1024, 512, 256, 128] | 80.4 | 75.7 | 90.3 | 87.2 | 88.3 | 89.4 | **87.0** | 87.4 | 2.3 |
| [2048, 1024, 512, 256] | 80.4 | **75.8** | **91.4** | **88.4** | **88.8** | 89.4 | 86.2 | **87.5** | **2.1** |
| [2048, 1024, 512, 256]* | 80.4 | 75.3 | 91.0 | 88.0 | 88.5 | 89.3 | 86.3 | 87.3 | 2.3 |
| [4096, 2048, 1024, 512] | **80.5** | 75.1 | 91.1 | 88.1 | 88.2 | **89.6** | 86.8 | 87.4 | 2.2 |

## D.3 Effect of the number of slots

For a fair comparison, we keep the number of experts constant ($E_1 \in \{4, 8, 12, 16\}$ and $E_2 \in \{8, 16, 24, 32\}$) and the number of groups fixed ($K \in \{2048, 1024, 512, 256\}$).

Table 20: Quantitative ablation study on the effect of the number of slots per expert ($S$) across the PANORAMA test sets. Performance is reported as DSC (%) for individual datasets and summarized via mDSC (%) and mHD95 (mm) for the aggregate average across all cohorts. The parameter $S$ represents the capacity of each expert to process tokens across the four encoder stages. The best and second-best results are **bolded** and underlined, respectively.

| Slots (S) | mDSC (%) ↑ | | | | | | | mDSC (%) ↑ | mHD95 (mm) ↓ |
| --- | --- | --- | --- | --- | --- | --- | --- | --- | --- |
| | NIH | MSD | | In-house | | Overall | | | |
| | Pancreas | PDAC | Pancreas | PDAC | Pancreas | PDAC | Pancreas | | |
| 1 | **93.0** | 46.6 | **88.6** | 55.9 | **82.2** | 50.1 | **88.5** | 76.2 | 5.4 |
| 2 | 92.5 | 49.7 | 88.0 | **61.5** | 82.0 | 54.2 | 88.0 | 77.2 | 5.0 |
| 4 | 92.9 | **51.2** | 88.4 | 60.8 | 81.7 | **54.8** | 88.3 | **77.5** | **4.8** |
| 8 | 92.8 | 46.4 | 88.3 | 60.6 | 81.3 | 51.8 | 88.1 | 76.5 | 5.3 |

Table 21: Quantitative ablation study on the effect of the number of slots per expert ($S$) on the AMOS-CT validation set. Performance is reported as DSC (%) for each of the 15 abdominal organs, with mDSC (%) and mHD95 (mm) representing the aggregate average. Organs include: Spleen (Sp), Right Kidney (RK), Left Kidney (LK), Gallbladder (GB), Esophagus (Es), Liver (Li), Stomach (St), Aorta (Ao), Inferior Vena Cava (IVC), Pancreas (Pa), Right Adrenal Gland (RAG), Left Adrenal Gland (LAG), Duodenum (Du), Bladder (Bl), and Prostate/Uterus (Pr/Ut). The parameter $S$ represents the capacity allocated to each expert for token processing across the four encoder stages. The best and second-best results are **bolded** and underlined, respectively.

| Slots (S) | Sp | RK | LK | GB | Es | Li | St | Ao | IVC | Pa | RAG | LAG | Du | Bl | Pr/Ut | mDSC (%) ↑ | mHD95 (mm) ↓ |
| --- | --- | --- | --- | --- | --- | --- | --- | --- | --- | --- | --- | --- | --- | --- | --- | --- | --- |
| 1 | 95.0 | 93.0 | 95.1 | 80.6 | 81.4 | 95.7 | 87.4 | 93.5 | 88.8 | **84.4** | **75.0** | 74.8 | 77.8 | 83.8 | 82.4 | 85.9 | 19.6 |
| 2 | 95.5 | 95.3 | 95.1 | 80.3 | 82.3 | 91.3 | 87.7 | 93.7 | 88.9 | 84.0 | 74.6 | 74.5 | **78.6** | 86.2 | 81.3 | 86.0 | 19.4 |
| 4 | **96.0** | 95.0 | 94.4 | **81.7** | 82.0 | 94.7 | **90.2** | **93.8** | 88.9 | 84.0 | 74.4 | 74.1 | 78.4 | 83.8 | **83.2** | **86.3** | **18.7** |
| 8 | 95.3 | **95.8** | **95.3** | 79.5 | 82.3 | **96.1** | 87.6 | 93.3 | 88.7 | 82.2 | 74.9 | **75.4** | 76.2 | **87.6** | 81.6 | 86.1 | 19.1 |

Table 22: Quantitative ablation study on the effect of the number of slots per expert ($S$) across 5-fold cross-validation on the FeTA 2022 dataset. Performance is reported as DSC (%) for each fetal brain tissue class and summarized via mDSC (%) and mHD95 (mm) for the aggregate average across all folds. Tissue classes include: external cerebrospinal fluid (eCSF), gray matter (GM), white matter (WM), ventricles (V), cerebellum (C), deep gray matter (dGM), and brainstem (B). The parameter $S$ defines the token processing capacity of each expert across the four encoder stages. The best and second-best results are **bolded** and underlined, respectively.

| Slots (S) | eCSF | GM | WM | V | C | dGM | B | mDSC (%) ↑ | mHD95 (mm) ↓ |
| --- | --- | --- | --- | --- | --- | --- | --- | --- | --- |
| 1 | 80.6 | 75.1 | 91.0 | 87.9 | 88.4 | 89.4 | 86.4 | 87.3 | 2.2 |
| 2 | **80.7** | 75.3 | 91.1 | 87.9 | 88.5 | **89.6** | 86.4 | 87.4 | 2.4 |
| 4 | 80.4 | **75.8** | **91.4** | **88.4** | **88.8** | 89.4 | 86.2 | **87.5** | **2.1** |
| 8 | 80.5 | 75.3 | 91.1 | 88.1 | 88.5 | 89.5 | **88.6** | 87.4 | 2.2 |

## D.4 Impact of the HoME layer

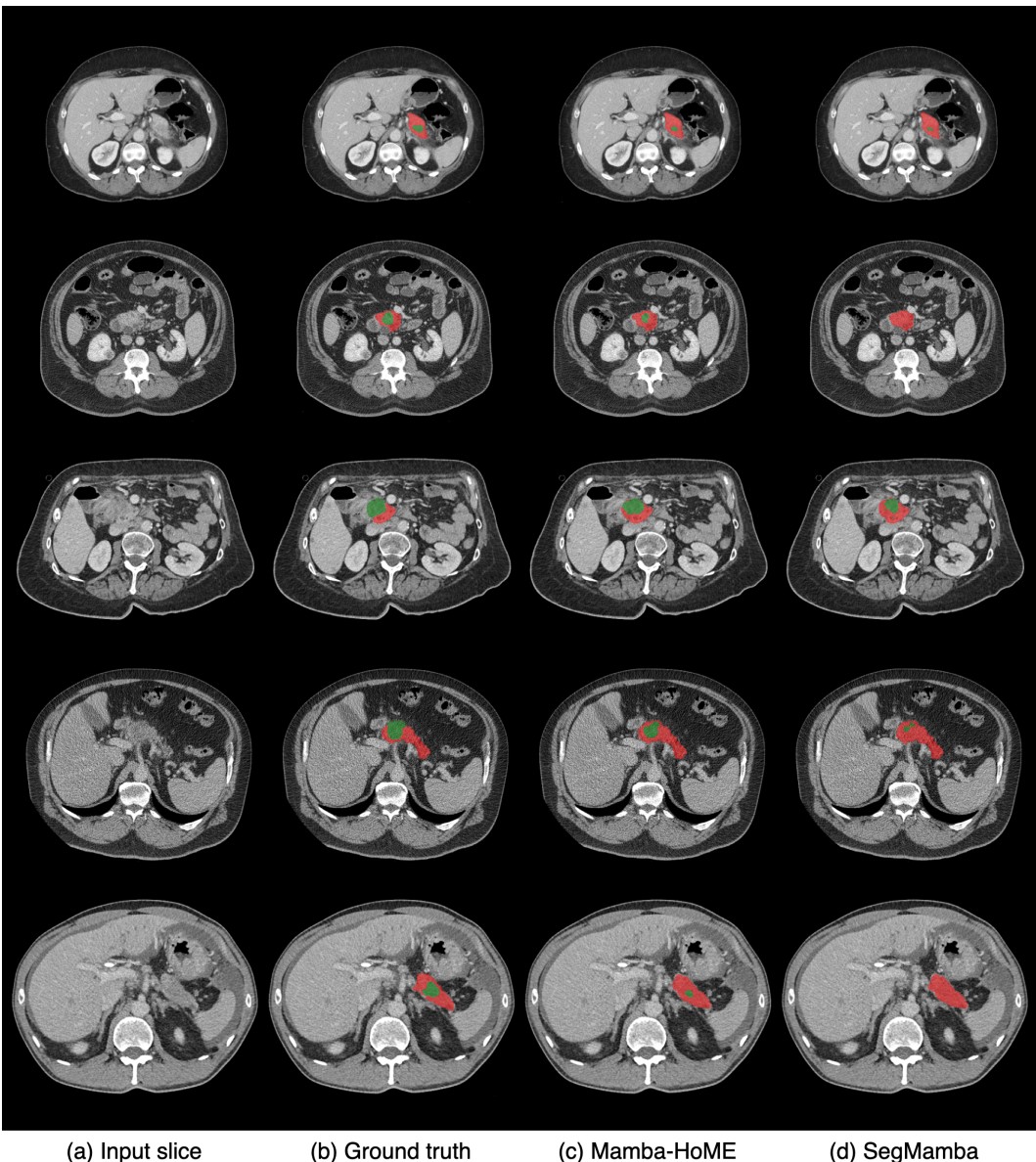

     (a) Input slice     (b) Ground truth     (c) Mamba-HoME     (d) SegMamba

Figure 4: Qualitative comparison of Mamba-HoME and SegMamba on abdominal CT scans from the PANORAMA test set. These images demonstrate the efficacy of integrating the HoME layer into the baseline SegMamba architecture. The green (PDAC) and red (pancreas) annotations highlight key segmentation differences, showing that Mamba-HoME is more robust in identifying anatomical structures across varying scales — from small tumors to larger organ boundaries. Notably, the Mamba-HoME results shown were achieved through training from scratch.

# E Qualitative results

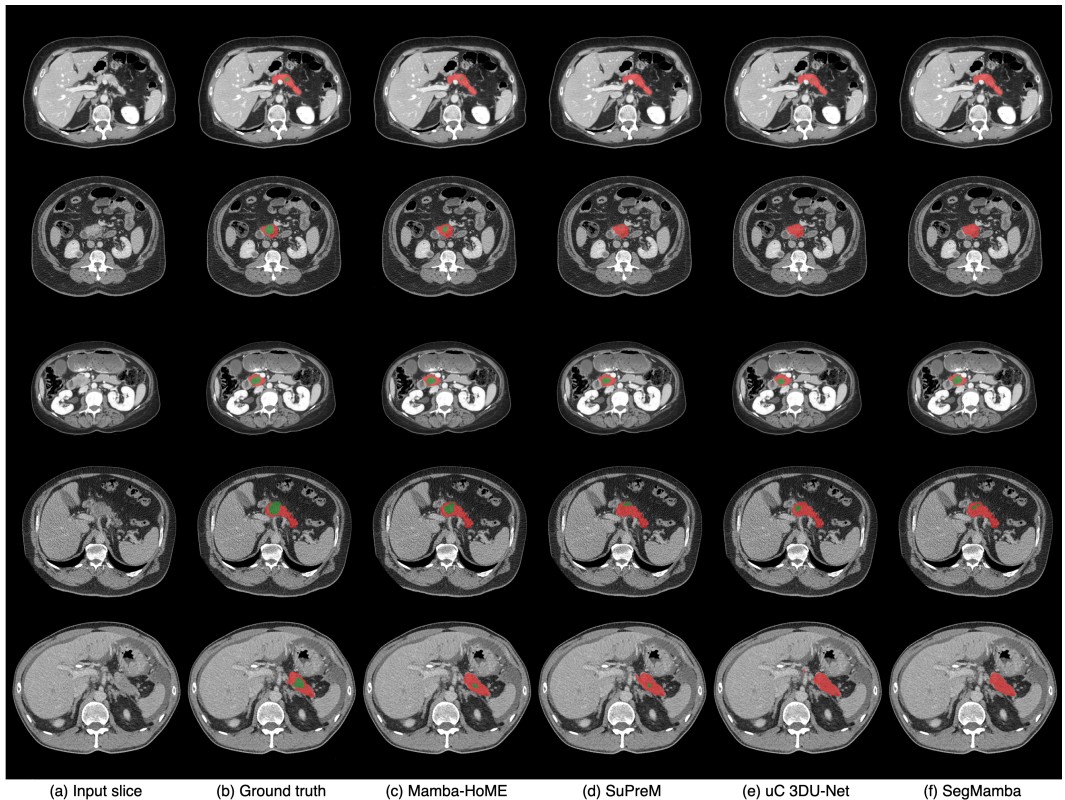

Figure 5: Qualitative segmentation results for PDAC (green) and the pancreas (red) provided by Mamba-HoME and the next three top-performing methods. The first three rows display cases from the MSD Pancreas dataset, while the last two rows show cases from the in-house dataset. Please note, we show Mamba-HoME results trained from scratch.

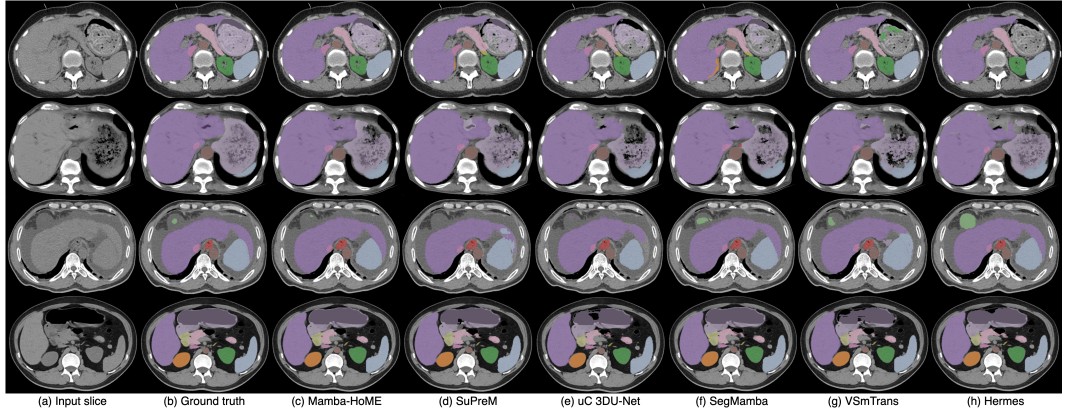

Figure 6: Qualitative segmentation results on the AMOS-CT validation set for Mamba-HoME (trained from scratch), SuPreM, uC 3DU-Net, SegMamba, VSmTrans, and Hermes. All models are trained on both the AMOS-CT and AMOS-MRI training datasets.

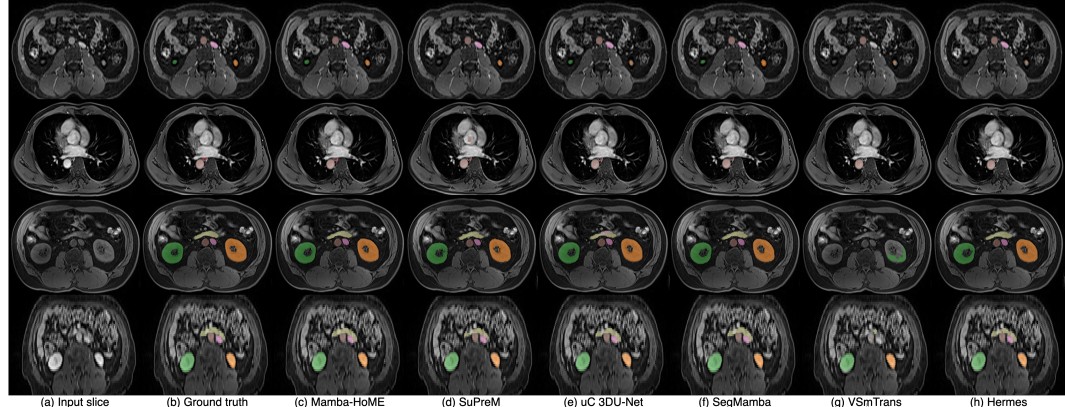

(a) Input slice  (b) Ground truth  (c) Mamba-HoME  (d) SuPreM  (e) uC 3DU-Net  (f) SegMamba  (g) VSmTrans  (h) Hermes

Figure 7: Qualitative comparison of segmentation performance on the AMOS-MRI validation set for six methods: Mamba-HoME (trained from scratch), SuPreM, uC 3DU-Net, SegMamba, VSmTrans, and Hermes. The models are trained on both training dataset of AMOS-CT and AMOS-MRI.

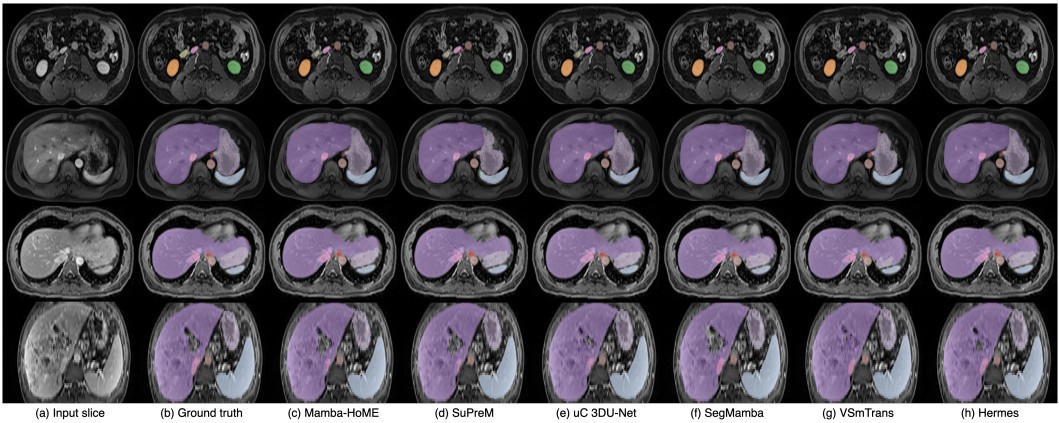

(a) Input slice  (b) Ground truth  (c) Mamba-HoME  (d) SuPreM  (e) uC 3DU-Net  (f) SegMamba  (g) VSmTrans  (h) Hermes

Figure 8: Qualitative segmentation results on the AMOS-MRI validation set for Mamba-HoME, SuPreM, uC 3DU-Net, SegMamba, VSmTrans, and Hermes. Models are first pre-trained on the AMOS-CT scans and subsequently fine-tuned on the AMOS-MRI training data.

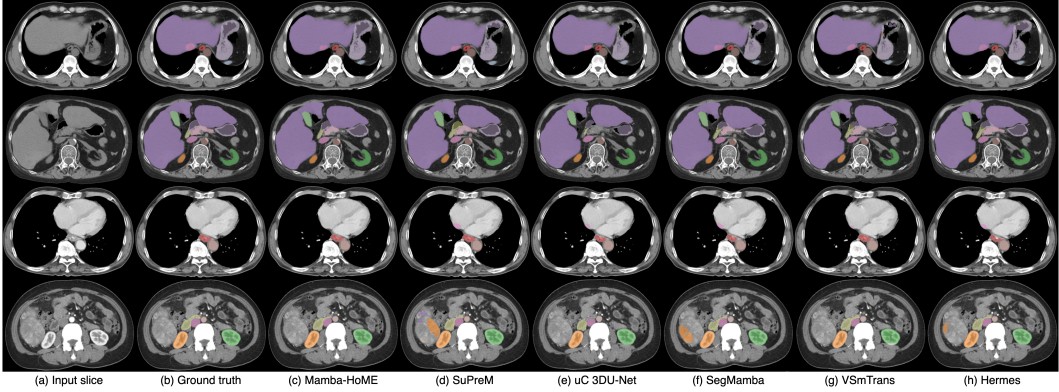

(a) Input slice  (b) Ground truth  (c) Mamba-HoME  (d) SuPreM  (e) uC 3DU-Net  (f) SegMamba  (g) VSmTrans  (h) Hermes

Figure 9: Qualitative segmentation results on the AMOS-CT validation set for Mamba-HoME, SuPreM, uC 3DU-Net, SegMamba, VSmTrans, and Hermes. Each model is trained only on the AMOS-CT training scans.

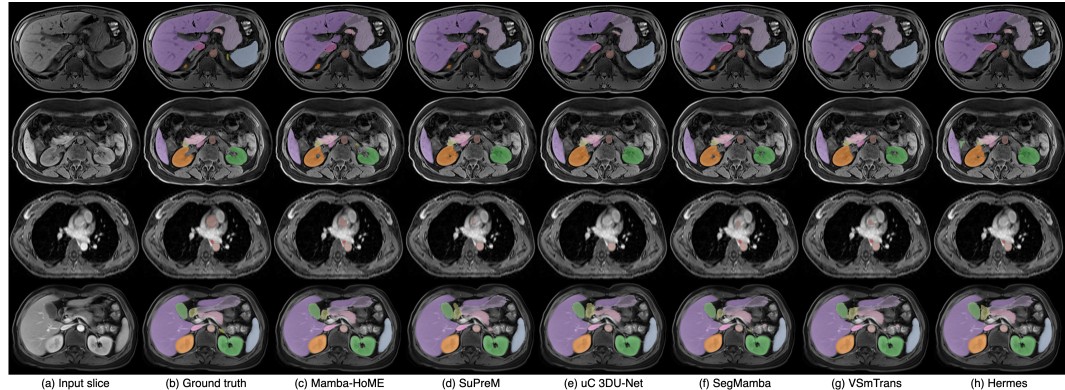

| (a) Input slice | (b) Ground truth | (c) Mamba-HoME | (d) SuPreM | (e) uC 3DU-Net | (f) SegMamba | (g) VSmTrans | (h) Hermes |

Figure 10: Qualitative segmentation results on the AMOS-MRI validation set for Mamba-HoME, SuPreM, uC 3DU-Net, SegMamba, VSmTrans, and Hermes. These models are trained solely on the AMOS-MRI training set.

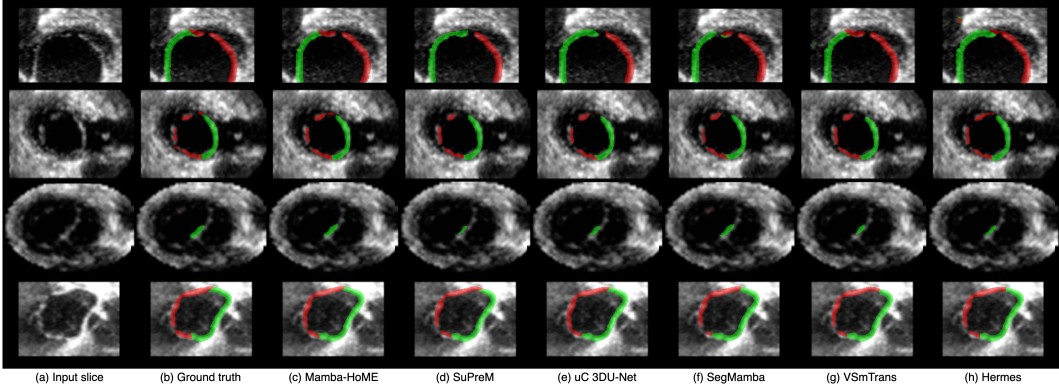

| (a) Input slice | (b) Ground truth | (c) Mamba-HoME | (d) SuPreM | (e) uC 3DU-Net | (f) SegMamba | (g) VSmTrans | (h) Hermes |

Figure 11: Qualitative segmentation results on the MVSeg test set for Mamba-HoME, SuPreM, uC 3DU-Net, SegMamba, VSmTrans, and Hermes.

# F   Generalizability analysis

Figure 12 provides an overview of the datasets utilized in the Mamba-HoME framework, encompassing CT, MRI, fetal MRI, and 3D ultrasound modalities. The figure presents voxel-wise ground truth labels across cross-modal and cross-anatomical domains. The first and third columns, as well as the second and fourth columns, display independent input slices and their corresponding ground truth segmentations for two representative cases. The framework, initially developed using CT and MRI scans, demonstrates consistent segmentation of abdominal organs such as the liver, spleen, and kidneys. The inclusion of fetal MRI and 3D ultrasound for fine-tuning further highlights the model's capacity to generalize across modalities with distinct anatomical features and imaging characteristics. This cross-modal and cross-anatomical representation emphasizes the versatility of the Mamba-HoME network in capturing variability across both imaging techniques and anatomical structures.

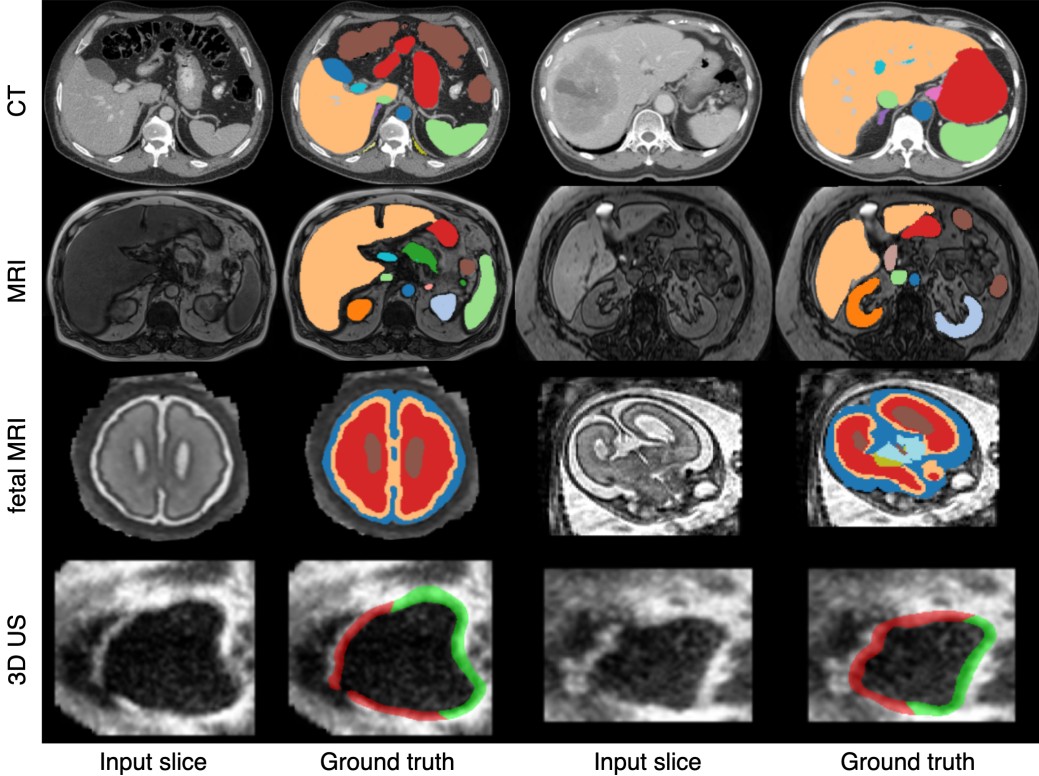

Figure 12: This figure illustrates the diversity of modalities used across the training pipeline. The model is pre-trained on CT scans (Row 1) and MRI scans (Row 2), then fine-tuned on downstream tasks involving fetal MRI (Row 3) and 3D ultrasound (Row 4). For two separate cases, Column 1 and Column 3 present the input slices, while Column 2 and Column 4 display the corresponding ground truth. This arrangement highlights the significant domain shift in feature representation, moving from the high-resolution anatomical structures of CT and MRI to the distinct imaging characteristics and noise profiles found in fetal MRI and 3D ultrasound.

We perform a comprehensive quantitative evaluation of the Mamba-HoME's generalizability across four distinct experimental protocols. First, we establish intra-modal baselines by training and evaluating the model independently on the AMOS-CT (see Table 23) and AMOS-MRI (see Table 24) datasets. Second, we investigate cross-modal transfer learning by pre-training the architecture on AMOS-CT followed by fine-tuning on AMOS-MRI, assessing the efficacy of knowledge transfer across disparate imaging physics (see Table 25). Finally, we conduct joint multi-modal training, where the network is trained simultaneously on both AMOS-CT and AMOS-MRI and evaluated across their respective validation sets (see Table 26) to determine the model's capacity for learning domain-invariant feature representations.

Table 23: Quantitative segmentation results on the AMOS-CT validation set. All models were trained exclusively on CT data. Performance is reported using DSC (%) for each of the 15 abdominal organs and mDSC (%) and mHD95 (mm) for the final average across all structures. Organs include: Spleen (Sp), Right Kidney (RK), Left Kidney (LK), Gallbladder (GB), Esophagus (Es), Liver (Li), Stomach (St), Aorta (Ao), Inferior Vena Cava (IVC), Pancreas (Pa), Right Adrenal Gland (RAG), Left Adrenal Gland (LAG), Duodenum (Du), Bladder (Bl), and Prostate/Uterus (Pr/Ut). The best results are **bolded**, while the second-best are underlined. Methods marked with (*) are initialized with pre-trained weights.

| Method | DSC (%) ↑ | | | | | | | | | | | | | | | mDSC (%) ↑ | mHD95 (mm) ↓ |
|---|---|---|---|---|---|---|---|---|---|---|---|---|---|---|---|---|---|
| | Sp | RK | LK | GB | Es | Li | St | Ao | IVC | Pa | RAG | LAG | Du | Bl | Pr/Ut | | |
| Hermes | 95.7 | 94.5 | 95.3 | 80.9 | 81.1 | 93.9 | 88.0 | 92.9 | 87.8 | 82.5 | 73.7 | 71.8 | 77.7 | 83.3 | 79.3 | 85.3 | **11.3** |
| Swin SMT | 95.4 | **95.6** | 95.1 | 76.3 | 81.3 | 92.2 | 87.4 | 93.7 | 88.6 | 84.1 | 74.4 | 73.4 | 78.3 | 86.6 | 82.6 | 85.7 | 47.2 |
| VSmTrans | 95.6 | 94.7 | 94.5 | 82.7 | 76.4 | 95.2 | 86.9 | 92.9 | 83.1 | 84.0 | 72.9 | **74.7** | 77.8 | 86.8 | 81.6 | 85.3 | 39.1 |
| SegMamba | 95.7 | 95.5 | 95.2 | 81.7 | 81.1 | 94.9 | 87.7 | 93.2 | 88.8 | 84.0 | **74.7** | 74.0 | 77.3 | 83.9 | 82.6 | 86.0 | 26.3 |
| uC 3DU-Net | 93.0 | 94.6 | 94.3 | 73.6 | 78.8 | 93.4 | 82.5 | 92.4 | 85.5 | 75.4 | 73.3 | 71.1 | 69.8 | 83.0 | 79.1 | 82.7 | 19.1 |
| Swin UNETR | 95.2 | 95.1 | 94.5 | 72.2 | 80.4 | 90.2 | 81.0 | 93.2 | 87.9 | 81.6 | 74.4 | 74.5 | 78.1 | 85.0 | 80.8 | 84.3 | 43.6 |
| SuPreM (*) | 95.6 | 95.0 | 94.4 | **86.2** | 79.4 | **97.1** | **91.2** | 93.0 | 88.7 | **85.3** | 69.3 | 68.8 | **81.3** | 86.0 | 78.9 | 86.0 | 16.5 |
| **Mamba-HoME** | **96.0** | 95.0 | 94.4 | 81.7 | 82.0 | 94.7 | 90.2 | **93.8** | **88.9** | 84.0 | 74.4 | 74.1 | 78.4 | 83.8 | **83.2** | 86.3 | 18.7 |
| **Mamba-HoME** (*) | **96.0** | 95.4 | **95.4** | 85.3 | **82.1** | 96.8 | **91.3** | 93.3 | 88.6 | 84.7 | 74.0 | 74.6 | 80.8 | **88.0** | 82.8 | **87.3** | 12.2 |

Table 24: Quantitative segmentation results on the AMOS-MRI validation set. All models were trained exclusively on MRI data. Performance is reported using DSC (%) for each of the 15 abdominal organs, with mDSC (%) and mHD95 (mm) representing the final average across all available structures. Organs include: Spleen (Sp), Right Kidney (RK), Left Kidney (LK), Gallbladder (GB), Esophagus (Es), Liver (Li), Stomach (St), Aorta (Ao), Inferior Vena Cava (IVC), Pancreas (Pa), Right Adrenal Gland (RAG), Left Adrenal Gland (LAG), Duodenum (Du), Bladder (Bl), and Prostate/Uterus (Pr/Ut). The best results are **bolded**, while the second-best are underlined. Methods marked with (*) are initialized with pre-trained weights, while NA indicates organs not present or labeled in this specific validation subset.

| Method | DSC (%) ↑ | | | | | | | | | | | | | | | mDSC (%) ↑ | mHD95 (mm) ↓ |
|---|---|---|---|---|---|---|---|---|---|---|---|---|---|---|---|---|---|
| | Sp | RK | LK | GB | Es | Li | St | Ao | IVC | Pa | RAG | LAG | Du | Bl | Pr/Ut | | |
| Hermes | 95.5 | 95.4 | 95.3 | 68.0 | **72.3** | 96.7 | 83.4 | 90.2 | 87.2 | 79.2 | 61.9 | 60.8 | 63.1 | NA | NA | 80.7 | 13.0 |
| Swin SMT | 92.8 | 94.2 | 93.7 | 56.5 | 0.0 | 95.8 | 81.1 | 88.6 | 83.2 | 79.4 | 0.0 | 0.0 | 56.3 | NA | NA | 63.2 | 22.5 |
| VSmTrans | 94.8 | 94.3 | 94.9 | 65.3 | 69.7 | 96.6 | 83.8 | 90.6 | 85.5 | 80.3 | 52.4 | 0.0 | 61.5 | NA | NA | 74.6 | 18.5 |
| SegMamba | 95.6 | 95.1 | 95.3 | 64.3 | 70.2 | 96.9 | 87.1 | **91.2** | 86.6 | 80.2 | 56.3 | 59.8 | 63.7 | NA | NA | 80.2 | 12.6 |
| uC 3DU-Net | 91.6 | 92.6 | 93.4 | 71.0 | 67.1 | 95.0 | 83.2 | 89.0 | 83.4 | 79.4 | 0.0 | 0.0 | 59.8 | NA | NA | 68.6 | 16.9 |
| Swin UNETR | 93.5 | 94.5 | 94.7 | 70.3 | 68.2 | 96.3 | 84.4 | 90.2 | 86.0 | 80.5 | 54.5 | 0.0 | 61.5 | NA | NA | 75.0 | 14.7 |
| SuPreM (*) | 93.6 | 94.6 | 94.0 | 71.2 | 67.3 | 95.9 | 83.7 | 89.3 | 83.8 | 79.8 | 0.0 | 0.0 | 60.6 | NA | NA | 70.3 | 18.2 |
| **Mamba-HoME** | 95.2 | 94.6 | 94.6 | 67.6 | 70.6 | 96.7 | 85.0 | 90.0 | **87.6** | 81.6 | **62.7** | 60.5 | 65.8 | NA | NA | 81.0 | 11.7 |
| **Mamba-HoME** (*) | **96.1** | **95.5** | **95.7** | **75.3** | **72.3** | **97.5** | **89.1** | 90.4 | 86.5 | **85.3** | 58.2 | **62.1** | **65.9** | NA | NA | **82.3** | **11.0** |

Table 25: Quantitative segmentation results on the AMOS-MRI validation set via cross-modal transfer (CT → MRI). Models were first pre-trained on the AMOS-CT dataset and subsequently fine-tuned on the AMOS-MRI training set prior to evaluation. Performance is reported using DSC (%) for each of the 15 abdominal organs, with mDSC (%) and mHD95 (mm) representing the aggregate average across available structures. Organs include: Spleen (Sp), Right Kidney (RK), Left Kidney (LK), Gallbladder (GB), Esophagus (Es), Liver (Li), Stomach (St), Aorta (Ao), Inferior Vena Cava (IVC), Pancreas (Pa), Right Adrenal Gland (RAG), Left Adrenal Gland (LAG), Duodenum (Du), Bladder (Bl), and Prostate/Uterus (Pr/Ut). The best results are **bolded**, while the second-best are underlined. Methods marked with (*) are initialized with pre-trained weights, while NA indicates organs not present or labeled in this specific validation subset.

| Method | Sp | RK | LK | GB | Es | Li | St | Ao | IVC | Pa | RAG | LAG | Du | Bl | Pr/Ut | mDSC (%) ↑ | mHD95 (mm) ↓ |
|---|---|---|---|---|---|---|---|---|---|---|---|---|---|---|---|---|---|
| Hermes | **96.6** | **95.8** | 96.0 | 77.7 | **78.3** | 97.7 | 90.4 | **92.5** | 89.9 | 86.0 | 65.2 | 63.5 | 72.0 | NA | NA | 84.8 | 8.2 |
| Swin SMT | 96.1 | 95.4 | 95.7 | 75.7 | 75.2 | 97.3 | 88.2 | 91.7 | 89.1 | 82.3 | 62.9 | 65.2 | 65.9 | NA | NA | 83.2 | 9.0 |
| VSmTrans | 96.2 | **95.8** | 95.7 | 79.9 | 74.1 | 87.5 | 90.5 | 91.2 | 89.5 | 85.4 | 64.2 | 66.8 | 70.8 | NA | NA | 84.5 | 14.1 |
| SegMamba | 94.7 | 95.0 | 95.0 | 67.4 | 69.9 | 96.3 | 84.3 | 90.5 | 84.7 | 80.8 | 57.0 | 50.5 | 60.6 | NA | NA | 79.0 | 8.2 |
| uC 3DU-Net | 96.1 | **95.8** | 95.8 | **80.1** | 76.8 | 97.3 | 88.5 | 91.8 | 89.1 | 84.3 | 63.8 | 68.4 | 71.0 | NA | NA | 84.5 | 9.1 |
| Swin UNETR | 96.1 | 95.6 | 95.8 | 78.9 | 72.4 | 97.2 | 89.2 | 91.0 | 88.9 | 85.1 | 65.9 | 69.3 | 68.6 | NA | NA | 84.2 | 13.3 |
| SuPreM (*) | 96.1 | 95.3 | 95.8 | 76.7 | 73.9 | 97.5 | 88.9 | 91.8 | 88.8 | 85.2 | 63.5 | 66.5 | 71.0 | NA | NA | 83.9 | 8.6 |
| **Mamba-HoME** | 96.4 | 95.5 | 95.9 | 77.3 | 76.2 | 97.5 | 90.9 | 91.4 | 89.4 | 86.4 | 64.5 | 66.7 | **72.2** | NA | NA | 84.8 | 8.1 |
| **Mamba-HoME** (*) | 96.5 | **95.8** | 96.1 | 79.5 | 77.0 | **97.8** | 91.2 | 92.4 | 90.4 | 85.4 | 63.9 | 66.5 | 72.0 | NA | NA | 85.0 | **8.0** |

Table 26: Quantitative segmentation results on the joint AMOS-CT and AMOS-MRI validation sets. Models were trained concurrently on both CT and MRI training sets and evaluated across both modalities. Performance is reported using DSC (%) for each of the 15 abdominal organs, while mDSC (%) and mHD95 (mm) represent the aggregate average across all structures. Organs include: Spleen (Sp), Right Kidney (RK), Left Kidney (LK), Gallbladder (GB), Esophagus (Es), Liver (Li), Stomach (St), Aorta (Ao), Inferior Vena Cava (IVC), Pancreas (Pa), Right Adrenal Gland (RAG), Left Adrenal Gland (LAG), Duodenum (Du), Bladder (Bl), and Prostate/Uterus (Pr/Ut). The best and second-best results are **bolded** and underlined, respectively. The best results are **bolded**, while the second-best are underlined. Methods marked with (*) are initialized with pre-trained weights.

| Method | Sp | RK | LK | GB | Es | Li | St | Ao | IVC | Pa | RAG | LAG | Du | Bl | Pr/Ut | mDSC (%) ↑ | mHD95 (mm) ↓ |
|---|---|---|---|---|---|---|---|---|---|---|---|---|---|---|---|---|---|
| Hermes | 94.5 | 93.7 | 93.3 | 77.8 | 77.6 | 93.8 | 87.2 | 92.5 | 87.7 | 82.0 | 71.7 | 70.0 | 72.2 | 83.3 | 79.3 | 82.9 | 7.3 |
| Swin SMT | 91.1 | 94.0 | 94.2 | 75.1 | 76.7 | 94.0 | 79.2 | 90.8 | 85.6 | 80.3 | 68.1 | 70.3 | 68.8 | 70.3 | 76.8 | 81.2 | 17.7 |
| VSmTrans | 85.0 | 87.1 | 89.1 | 71.7 | 73.9 | 86.3 | 74.5 | 87.5 | 79.2 | 74.5 | 64.3 | 67.2 | 67.3 | 82.6 | 80.8 | 78.0 | 15.2 |
| SegMamba | 94.9 | 94.0 | 93.5 | 76.7 | 79.7 | 94.5 | 89.1 | **93.1** | **88.3** | 83.1 | 68.5 | 72.5 | 76.0 | 83.4 | 82.3 | 84.7 | 7.5 |
| uC 3DU-Net | 95.0 | 94.5 | 93.8 | 76.3 | 78.8 | 95.3 | 86.9 | 92.7 | 87.6 | 81.9 | 71.7 | 70.7 | 74.0 | 83.2 | 76.7 | 84.1 | 12.2 |
| Swin UNETR | 92.1 | 94.8 | 94.6 | 78.1 | 76.8 | 95.1 | 86.5 | 91.8 | 84.7 | 82.8 | 69.1 | 71.0 | 74.6 | 83.1 | 81.5 | 83.8 | 12.4 |
| SuPreM (*) | 92.8 | 92.6 | 94.1 | 78.5 | 78.1 | 93.2 | 84.4 | 92.3 | 84.5 | 82.3 | 67.8 | 71.9 | 74.2 | 84.4 | 81.2 | 83.5 | 11.8 |
| **Mamba-HoME** | 95.2 | 95.1 | 94.6 | 79.3 | 79.9 | 95.4 | 89.1 | 92.8 | 88.2 | 82.7 | 71.9 | 72.7 | 75.5 | 82.7 | 80.5 | 85.1 | 7.4 |
| **Mamba-HoME** (*) | 95.0 | 95.3 | 95.6 | 82.7 | 80.0 | 96.3 | 90.4 | 93.0 | 87.8 | 84.5 | 72.1 | 72.9 | 77.7 | 89.0 | 83.2 | 86.4 | 7.2 |

# G Impact statement

This work introduces Hierarchical Soft Mixture-of-Experts (HoME), a novel architecture for efficient and accurate 3D medical image segmentation. By integrating a two-level mixture-of-experts routing mechanism with Mamba-based Selective State-Space Models, our method significantly advances long-context modeling for volumetric data. HoME is designed to address key challenges in medical imaging, namely, modeling local-to-global spatial hierarchies, handling modality diversity (CT, MRI, US), and achieving scalability for high-resolution 3D inputs. Our proposed Mamba-HoME architecture demonstrates strong generalization and outperforms state-of-the-art models across public and in-house datasets, while being memory and compute efficient.

Beyond medical imaging, the architectural principles introduced, specifically the hierarchical token routing and the integration of local and global context processing, are applicable to other domains dealing with structured, hierarchical data under resource constraints. These include scientific computing, robotics, and spatiotemporal analysis in environmental or geospatial datasets.

Ethically, this work supports equitable healthcare by enabling accurate segmentation with reduced computational requirements, which is crucial for deployment in low-resource settings. We use publicly available datasets and provide open-source code to ensure reproducibility and accessibility for the broader community. No personally identifiable information or sensitive patient data is used. Future extensions could include further robustness to distributional shifts in medical data and broader clinical evaluation.

