# OpenReview forum: "Mamba Goes HoME: Hierarchical Soft Mixture-of-Experts for 3D Medical Image Segmentation"
_NeurIPS.cc/2025/Conference — NeurIPS 2025 poster_

### Official Review · Reviewer_k1E5 · 2025-06-20

**Clarity:** 2
**Significance:** 3
**Originality:** 3
**Rating:** 4
**Confidence:** 4

**Summary:**

This paper proposes a new framework, Mamba-HoME, for 3D medical image segmentation. It integrates Hierarchical Soft Mixture-of-Experts (HoME) into the Mamba model, which efficiently handles long-range dependencies and local-to-global feature extraction. HoME is designed with two levels of token-routing experts: one for local feature extraction and the other for global context aggregation. The authors demonstrate that Mamba-HoME outperforms existing state-of-the-art methods in segmentation tasks across multiple 3D imaging modalities, exhibiting improved accuracy and efficiency.

**Questions:**

- Could the authors provide further insights into how the model performs with extremely noisy or low-resolution data, particularly in clinical environments where such data is more common?
- How do the number of experts and the group sizes affect computational efficiency during inference, especially with higher-resolution datasets?

**Ethical Concerns:**

["NO or VERY MINOR ethics concerns only"]

**Final Justification:**

The motivation is good and the methods seem solid.

**Limitations:**

- Scalability: The paper acknowledges the limitation in terms of scalability for very large-scale datasets, which remains an unexplored area for future work.
- The paper mentions that the model is pre-trained using supervised learning on multimodal datasets. However, its behavior under self-supervised learning, especially on large unannotated datasets, remains unexplored.
- The model is not tested across a broader range of medical image types, limiting its potential for deployment across all clinical applications.

**Quality:**

3

**Strengths And Weaknesses:**

## Strengths
- Quality: The paper presents a novel and well-defined methodology, Mamba-HoME, integrating advanced techniques such as Mixture-of-Experts (MoE) and state-space models, and addresses important challenges in 3D medical image segmentation.
- Clarity: The paper clearly explains the architecture and methodology, providing adequate details on the hierarchical expert structure and token routing process.
- Significance: The model offers significant improvements over existing models in segmentation accuracy across various modalities, making it relevant for practical medical applications.
- Originality: The integration of hierarchical MoE layers within a state-space model framework for 3D medical segmentation is a unique and novel approach.

## Weaknesses
- Ablation study on the specific choices of parameters for the MoE layers (e.g., number of experts, group sizes) could benefit from further discussion on how they influence the model’s performance.
- The paper does not explore the scalability of the method to very large datasets (e.g., >10,000 scans), which limits understanding of the model’s real-world applicability to massive medical image datasets.

---

> ### Author Rebuttal · Authors · 2025-07-30
>
> > W1: Ablation study on the specific choices of parameters for the MoE layers (e.g., number of experts, group sizes) could benefit from further discussion on how they influence the model’s performance.
>
> We agree with the Reviewer that the ablation study should be accompanied by a more thorough discussion of the results. In response, we conducted additional experiments (see detailed results in response to Question 2 (Q2)) and will include these findings, along with an expanded analysis, in the final version of the paper.
>
> > W2: The paper does not explore the scalability of the method to very large datasets (e.g., >10,000 scans), which limits understanding of the model’s real-world applicability to massive medical image datasets.
>
> Following this and other Reviewer’s comments, we have now trained the model on the large AbdomenAtlas 1.1 [1] dataset, which consists of 9,262 CT scans in total. Of these, we used 8,242 for training and 1,020 for inference time and memory analysis. For training, we used 128 × 128 × 128 patches and a batch size = 2.
>
> According to the Table, Mamba-HoME has a training time of around 3,930 seconds (65.5 minutes), 0.47 s per scan, comparable to other large models, but even shorter than the lightweight uC 3DU-Net (0.59 s per scan). The average inference time for Mamba-HoME is approximately 2 seconds per scan, with a GPU memory requirement of 10.2 GB, which represents a good trade-off between speed and memory usage compared to other state-of-the-art methods.
>
> This new Table will be incorporated in the final version of the paper.
>
> | Method | Parameters (M) | Training time (s) | Inference time (s) | GPU memory (GB) |
> |:----------------:|:--------------:|:-----------------:|:------------------:|:---------------:|
> | uC 3DU-Net |      21.7      |       0.59        |        1.57        |      12.4       |
> | SuPreM |     62.2      |       0.40        |        2.32        |      15.8       |
> | Swin SMT |    170.8      |       0.42        |        2.09        |      14.6       |
> | Swin UNETR |      72.8      |       0.40        |        2.32        |      15.8       |
> | Hermes |      44.5      |       0.44        |        2.27        |      16.2       |
> | VSmTrans |      47.6      |       0.60        |        2.48        |       9.8       |
> | SegMamba-B |      66.8      |       0.38        |        1.52        |       9.0       |
> | SegMamba-L |     176.5      |       0.49        |        2.53        |      12.0       |
> | Mamba-HoME |     170.5      |       0.47        |        2.01        |      10.2       |
>
> We also note that the training time and inference times on large datasets generally scale linearly with the number of samples. Therefore, we do not anticipate that application of our model to even larger datasets would incur runtime issues.
>
> Finally, we note that if training on very large datasets (containing thousands of samples) would benefit from further scaling the model to larger parameter numbers, it is technically very easy to scale due to the scalability of MoE [2]. Specifically, to scale the model, we would only require to increase the number of experts.
>
> > Q1: Could the authors provide further insights into how the model performs with extremely noisy or low-resolution data, particularly in clinical environments where such data is more common?
>
> We appreciate the reviewer’s thoughtful question regarding model performance under noisy or low-resolution conditions, which are indeed common in clinical environments. To address this, we note that our experiments already include evaluations on four diverse 3D clinical imaging modalities: CT, MRI, fetal brain MRI, and 3D ultrasound. Both fetal brain MRI (FeTA) and 3D ultrasound (MVSeg) are particularly challenging due to motion artifacts, low signal-to-noise ratio, and limited spatial resolution. Despite these challenges, our model demonstrated strong and consistent performance across all modalities, indicating robustness to both low-quality and heterogeneous input data, as shown in Tables 3 and 4.
>
> > Q2: How do the number of experts and the group sizes affect computational efficiency during inference, especially with higher-resolution datasets?
>
> We are grateful for this inspiring suggestion! We performed additional experiments using different numbers of experts and group sizes to show how they affect computational efficiency during inference, especially with higher-resolution datasets. In this case, we use the AMOS-CT dataset (Each CT scan consists of between 74 and 353 slices with a mean of 153. The mean voxel spatial resolution is 0.68 × 0.68 × 3.86 mm³), and we also selected the 100 largest CT scans from the AbdomenAtlas 1.1 dataset (Each CT scan consists of between 786 and 1,440 slices with a mean of 1,156. The mean voxel spatial resolution is 0.82 × 0.82 × 0.79 mm³).
>
> We varied the expert configurations and observed a gradual increase in both inference time and GPU memory usage as the number of experts increased. These results show that while more experts lead to higher specialization, the increase in inference time and memory usage remains moderate and predictable, confirming scalability. E1 and E2 indicate the number of experts in the first and the second level of HoME, respectively. In this experiment, we used K = [2048, 1024, 512, 256] groups as the default.
>
>
> | E1 | E2 | Inference time AMOS-CT (s) | Inference time 100 high-dimensional (s) | GPU memory AMOS-CT (GB) | GPU memory 100 high-dimensional (GB) |
> |:---------------:|:---------------:|:--------------------------:|:-----------------------:|:----------------------:|:-------------------:|
> | [4, 8, 12, 16] | [8, 16, 24, 32] | 4.70 | 17.10 | 11.2 | 15.1 |
> | [8, 16, 24, 32] | [8, 16, 24, 32] | 5.19 | 19.90 | 12.3 | 16.2 |
> | [8, 16, 24, 48] | [16, 32, 48, 96] | 5.53 | 22.67 | 13.0 | 16.9 |
> | [16, 32, 48, 64] | [16, 32, 48, 64] | 6.27 | 23.55 | 13.1 | 17.0 |
>
> We also tested various group sizes to study how granularity affects efficiency.
> This trade-off suggests that larger group sizes enhance efficiency, particularly for large-volume data, while smaller groups offer more flexibility with a slight cost.
>
> In this experiment, we use E1 = E2 = [4, 8, 12, 16] as the default, except **, where E1 = [4, 8, 12, 16], and E2 = [8, 16, 24, 32].
>
> | Group size (K) | Inference time AMOS-CT (s) | Inference time 100 high-dimensional (s) | GPU memory AMOS-CT (GB) | GPU memory 100 high-dimensional (GB) |
> |:--------------------------:|:---------------------------:|:------------------------:|:-----------------------:|:--------------------:|
> | [1024, 512, 256, 128] | 4.85 | 17.60 | 11.0 | 14.9 |
> | [2048, 1024, 512, 256]** | 4.70 | 17.10 | 11.2 | 15.1 |
> | [2048, 1024, 512, 256] | 4.60 | 17.00 | 11.0 | 14.9  |
> | [4096, 2048, 1024, 512] | 4.48 | 15.90 | 11.0 | 14.9 |
>
> Our findings indicate that increasing the number of experts results in a moderate and predictable increase in inference time and GPU memory usage. Even at the most demanding configuration ([16, 32, 48, 64] experts), inference remains practical, 6.27 seconds for AMOS-CT and 23.55 seconds for the largest CT volumes. These are acceptable runtimes given the scale of the input data, particularly for volumetric scans comprising over 1,000 slices. Note that, in practical 3D image analysis, segmentation is performed after image acquisition, and there is no need for real-time inference.
>
> Importantly, the high-dimensional CT scans used here represent extremely large cases such as whole-body scans, which are rare in clinical practice. In contrast, datasets like AMOS-CT, with an average of ~150 slices, reflect typical diagnostic studies used in routine clinical workflows (e.g., abdominal or pelvic CTs). For such cases, inference times remain around 4.7–6.3 seconds, and memory consumption is well within the capabilities of standard high-end GPUs.
>
> Additionally, varying the group sizes demonstrates that coarser groupings improve computational efficiency, especially for large-volume scans, with no detrimental impact on memory usage.
>
> > L1: Scalability: The paper acknowledges the limitation in terms of scalability for very large-scale datasets, which remains an unexplored area for future work.
>
> Indeed, as noted in the paper, our current experiments were conducted on datasets of moderate size to enable controlled evaluation and thorough analysis. Applying our method to very large-scale datasets (e.g., hundreds of millions of examples) remains an important direction for future work.
>
> > L2: The paper mentions that the model is pre-trained using supervised learning on multimodal datasets. However, its behavior under self-supervised learning, especially on large unannotated datasets, remains unexplored.
>
> We agree that exploring self-supervised pre-training on large unannotated datasets is an important future direction. However, our choice to adopt supervised pre-training on a smaller multimodal dataset was motivated by the findings of Li et al. [1], who demonstrate that supervised 3D models trained on relatively small datasets can transfer more effectively to downstream medical imaging tasks than models pre-trained in a self-supervised manner on large-scale data. Our focus in this work was to leverage these insights to maximize downstream performance under limited compute and data conditions.
>
> > L3: The model is not tested across a broader range of medical image types, limiting its potential for deployment across all clinical applications.
>
> We would like to clarify that we have evaluated our method on three of the most widely used and diverse 3D medical imaging modalities: CT, MRI, and ultrasound. These modalities cover a broad spectrum of imaging characteristics, including differences in contrast and textures, high and low resolutions, noise properties, and anatomical focus.
>
> **References**
>
> [1] Li et al. "How Well Do Supervised 3D Models Transfer to Medical Imaging Tasks?." ICLR, 2024
>
> [2] Puigcerver et al. "From Sparse to Soft Mixtures of Experts." ICLR, 2024

---

> > ### Comment · Reviewer_k1E5 · 2025-08-05
> > **Brief response**
> >
> > Thank you for your clarification. I stand by my original positive assessment.

---

> > > ### Author Response · Authors · 2025-08-05
> > > **Thank you**
> > >
> > > Dear Reviewer,
> > >
> > > We sincerely thank you for your time and positive feedback.
> > >
> > > Best regards,
> > >
> > > Authors

---

### Official Review · Reviewer_PbeP · 2025-07-02

**Clarity:** 3
**Significance:** 2
**Originality:** 2
**Rating:** 4
**Confidence:** 4

**Summary:**

The paper introduces the Hierarchical Soft Mixture-of-Experts (HoME), a novel two-level token-routing mechanism designed for efficient 3D medical image segmentation. It builds on the Mamba state-space model (SSM) to combine local and global context modeling effectively. The proposed method demonstrates robust cross-modality generalization and significant advancements in segmentation performance, particularly in handling hierarchical structures in 3D medical data. These contributions aim to address the challenges of scalability and generalization in 3D medical image analysis.

**Questions:**

1.	Include more detailed analysis of memory usage and training/inference times for large datasets.

2.	Provide additional visual or textual explanations of the HoME layer for readers less familiar with the technical details.

3.	Discuss potential latency concerns in real-world applications of the model.

**Ethical Concerns:**

["NO or VERY MINOR ethics concerns only"]

**Final Justification:**

After reading the rebuttal and other reviews, I think the paper does have some merits but also have some limitations. So I choose to keep my originial rating.

**Limitations:**

Yes

**Quality:**

3

**Strengths And Weaknesses:**

Strengths

1.	The methodology is robust and well-executed. The integration of the Hierarchical Soft Mixture-of-Experts (HoME) with the Mamba state-space model (SSM) addresses the need for efficient local-to-global feature modeling in 3D medical image segmentation.

2.	Extensive experimental results validate the proposed model's performance across diverse datasets and imaging modalities (CT, MRI, Ultrasound), demonstrating both accuracy improvements and computational efficiency.

3.	The ablation studies provide a thorough exploration of the impact of architectural and hyperparameter choices, strengthening the validity of the conclusions.

Weaknesses

1.	The model has 170.5M parameters, which is larger than the lightweight model (e.g. uC 3DU-Net is only 21.7M), and is about 30% slower than the baseline SegMamba. The accuracy improvement comes at the expense of speed. While the paper demonstrates improvements in computational efficiency, it lacks detailed discussions of memory constraints and training times for very large-scale datasets, which might impact its scalability. The significance of the computational trade-offs (e.g., slightly slower inference speeds compared to simpler baselines) is not fully analyzed in the context of real-world clinical workflows, where latency may be a critical factor.

2.	While the integration of state-space models and hierarchical MoE is novel, the core ideas build upon well-established frameworks (e.g., SSMs and MoEs). So the novelty is limited.

---

> ### Author Rebuttal · Authors · 2025-07-30
>
> > W1: The model has 170.5M parameters, which is larger than the lightweight model (e.g. uC 3DU-Net is only 21.7M), and is about 30% slower than the baseline SegMamba. The accuracy improvement comes at the expense of speed. While the paper demonstrates improvements in computational efficiency, it lacks detailed discussions of memory constraints and training times for very large-scale datasets, which might impact its scalability. The significance of the computational trade-offs (e.g., slightly slower inference speeds compared to simpler baselines) is not fully analyzed in the context of real-world clinical workflows, where latency may be a critical factor.
>
>
> We are grateful for Reviewer’s comments and address each of them below.
> Regarding accuracy improvement at the expense of speed. In real-world clinical 3D analysis applications, real-time inference speed is generally not critical, as segmentation is typically performed after image acquisition and reconstruction. In such settings, even several seconds of computation are considered acceptable.
>
> We also performed additional experiments and demonstrated that when SegMamba is scaled to the same number of parameters (SegMamba-L), it is slower and requires more GPU memory compared to our proposed method, Mamba-HoME. We ran the model on 1,020 CT scans from the AbdomenAtlas 1.1 [1] validation set to analyze mean inference time per scan and memory usage.
>
> | Method      | Inference time (s) | GPU memory (GB) |
> |-------------|:------------------:|:--------------:|
> | SegMamba-B  |       1.50         |      9.0       |
> | SegMamba-L  |       2.53         |      12.0      |
> | Mamba-HoME  |       2.01         |      10.2      |
>
>
> Regarding training times and memory constraints,  following Reviewer's suggestion we performed additional experiments on large scans, which are outlined in the answer to Q1 below. In general, training and inference times for large datasets scale linearly with the number of CT scans; therefore, the method should be scalable to larger datasets.
>
> Following the Reviewer’s insightful comment about performance on large datasets, we have now added an additional analysis in which we evaluate inference time and GPU memory usage for a set of high-resolution CT scans. For this experiment, we selected the 100 scans with the highest resolution from the AbdomenAtlas 1.1 dataset [1].
>
> Each CT scan consists of between 786 and 1,440 slices with a mean of 1,156. The mean voxel spatial resolution is 0.82 × 0.82 × 0.79 mm³.
>
> The inference times obtained by Mamba-HoME even for these exceptionally large scans are perfectly suitable for implementation in clinical settings, where, as mentioned, segmentation is performed not in real-time but after image acquisition.
>
> | Method        | Parameters (M) | Inference time (s) | GPU memory (GB) |
> |------------------|:------------------:|:----------------------:|:------------------:|
> | uC 3DU-Net        |       21.7         |         13.4           |        17.5        |
> | SuPreM            |       62.2         |         19.1           |        21.0        |
> | Swin SMT          |      170.8         |         15.2           |        19.7        |
> | Swin UNETR        |       72.8         |         19.2           |        21.1        |
> | Hermes            |       44.5         |         21.5           |        17.1        |
> | VSmTrans          |       47.6         |         19.9           |        14.9        |
> | SegMamba-B        |       66.8         |         14.5           |        14.0        |
> | SegMamba-L        |      176.5         |         21.5           |        17.1        |
> | Mamba-HoME        |      170.5         |         17.7           |        15.1        |
>
>
> > W2: While the integration of state-space models and hierarchical MoE is novel, the core ideas build upon well-established frameworks (e.g., SSMs and MoEs). So the novelty is limited.
>
> To the best of our knowledge, Mamba-HoME is the first model to provide an architectural framework that combines SSMs and hierarchical MoEs for the task of efficient 3D image segmentation.
>
>
> > Q1: Include more detailed analysis of memory usage and training/inference times for large datasets.
>
>
> To address this comment, we have now trained the model on the large AbdomenAtlas 1.1 dataset, which consists of 9,262 CT scans in total. Of these, we used 8,242 for training and 1,020 for inference time and memory analysis. For training, we used 128 × 128 × 128 patches and a batch size = 2.
> According to the Table below, Mamba-HoME has a training time of around 0.47 seconds per scan, comparable to other large models, but even shorter than the lightweight uC 3DU-Net.
>
> The average inference time for Mamba-HoME is approximately 2 seconds per scan, with a GPU memory requirement of 10.2 GB, which represents a good trade-off between speed and memory usage compared to other state-of-the-art methods.
>
> This new Table will be incorporated in the final version of the paper.
>
>
> |      Method      | Parameters (M) | Training time (s) | Inference time (s) | GPU memory (GB) |
> |:----------------:|:--------------:|:-----------------:|:------------------:|:---------------:|
> |   uC 3DU-Net     |      21.7      |       0.59        |        1.57        |      12.4       |
> |     SuPreM       |      62.2      |       0.40        |        2.32        |      15.8       |
> |    Swin SMT      |     170.8      |       0.42        |        2.09        |      14.6       |
> |   Swin UNETR     |      72.8      |       0.40        |        2.32        |      15.8       |
> |     Hermes       |      44.5      |       0.44        |        2.27        |      16.2       |
> |    VSmTrans      |      47.6      |       0.60        |        2.48        |       9.8       |
> |  SegMamba-B      |      66.8      |       0.38        |        1.52        |       9.0       |
> |  SegMamba-L      |     176.5      |       0.49        |        2.53        |      12.0       |
> |  Mamba-HoME      |     170.5      |       0.47        |        2.01        |      10.2       |
>
>
> > Q2: Provide additional visual or textual explanations of the HoME layer for readers less familiar with the technical details.
>
> The visualization of the HoME layer will be included in the final version of the paper.
>
> > Q3: Discuss potential latency concerns in real-world applications of the model.
>
> We appreciate the Reviewer’s attention to practical deployment considerations.
> In real-world clinical 3D imaging workflows, latency is typically not a limiting factor, as segmentation is performed after image acquisition and reconstruction, not in real time. This is particularly true in applications such as radiology, oncology planning, and surgical navigation, where segmentation is part of the offline post-processing pipeline. Even when segmentation is triggered during live image review by a radiologist or oncologist, an inference time of approximately 2 seconds is negligible compared to the overall image examination process, which typically takes at least several minutes.
>
> Furthermore, the hardware requirements for inference are modest and can be satisfied by a standard workstation or a modern laptop equipped with a dedicated GPU. That said, we acknowledge that ultra-low-latency inference may be necessary for certain time-sensitive applications, such as intraoperative guidance or robotic-assisted procedures. We will include these clarifications in the final version of the paper.
>
>
> **References**
>
> [1] Li et al. "Abdomenatlas: A large-scale, detailed-annotated, & multi-center dataset for efficient transfer learning and open algorithmic benchmarking." Medical Image Analysis, 2024

---

> > ### Comment · Reviewer_PbeP · 2025-08-06
> > **Response**
> >
> > Thank the authors for providing the detailed response. Most of my concerns are well addressed. I still keep my original positive assessment.

---

> > > ### Author Response · Authors · 2025-08-06
> > > **Thank you**
> > >
> > > Dear Reviewer,
> > >
> > > We sincerely thank you for your time and positive feedback.
> > >
> > > Best regards,
> > >
> > > Authors

---

### Official Review · Reviewer_SpNP · 2025-07-03

**Clarity:** 4
**Significance:** 3
**Originality:** 3
**Rating:** 4
**Confidence:** 5

**Summary:**

This author proposed a Mamba-HoME, a novel 3D medical image segmentation architecture combining Mamba state-space models with a Hierarchical Soft Mixture-of-Experts (HoME) layer. HoME uses two-level token routing to capture local-to-global dependencies efficiently. Evaluations across CT, MRI, and ultrasound datasets show state-of-the-art segmentation performance with favorable computational efficiency and generalizability.

**Questions:**

1. Some sentences in Introduction are very long—could you break them up for clarity?
2. Please  standardize font sizes across all tables for consistency?
3. How much improvement does HoME provide over plain Mamba when controlling for parameter count?
4. What is the runtime and GPU memory footprint on a standard high-resolution volume?
5. Did you test cross-institution generalization or domain shift robustness?
6. Is there evidence that the second-level global routing meaningfully improves performance?
7. Are the observed improvements statistically significant on all datasets?

**Ethical Concerns:**

["NO or VERY MINOR ethics concerns only"]

**Final Justification:**

Thank the authors for addressing my comments. Most of my concerns are well addressed. I still keep my original positive assessment.

**Limitations:**

Yes

**Quality:**

3

**Strengths And Weaknesses:**

Strengths:
The approach innovatively integrates SSMs with hierarchical expert routing, addressing the scalability issues of transformers for large 3D volumes. Extensive experiments across five datasets convincingly demonstrate consistent improvements in Dice score and boundary accuracy. The ablations systematically explore design choices (number of experts, group sizes, slots). The inclusion of pretraining and cross-modal transfer further strengthens the claim of generalizability.

Weakness:
Performance gains over prior SSM methods (e.g., SegMamba) are modest (~1–2% Dice). The complexity of HoME increases parameter count and inference time (~30% slower). No user studies or clinical validation are provided. Some claims about “unprecedented memory and compute efficiency” may be overstated since inference speed lags behind simpler baselines. Details about dataset preprocessing and training schedules are scattered across appendix sections.

---

> ### Author Rebuttal · Authors · 2025-07-30
>
> > W1: Performance gains over prior SSM methods (e.g., SegMamba) are modest (1–2% Dice). The complexity of HoME increases parameter count and inference time (30% slower).
>
> We are grateful for this comment. In Figures 2 and 6–12 (see Supplementary Material), we illustrate examples of clear qualitative improvements, particularly in the delineation of small or hard-to-detect structures such as pancreatic tumors. These improvements are not fully captured by modest gains in aggregate metrics like DSC.
> Additionally, Table 1 and Table 11 (see Supplementary Material) demonstrate a significant improvement in the segmentation of small objects, such as PDAC tumors.
> Indeed, as the Reviewer points out, Mamba-HoME has 30 percentage points lower inference speed than the baseline SegMamba. Still, in practical 3D image analysis, segmentation is performed after image acquisition, and there is no need for real-time inference. Therefore, improving the accuracy over competitor models is far more important and justifies a decrease in speed.
> On top of that, our new comparison demonstrated that inference is much faster for Mamba-HoME (2.01 s/scan) than for SegMamba scaled to a similar parameter count (2.53 s/scan). Please refer to our answer to question **Q1 in reply to Reviewer PbeP** for full details and the relevant table. At the same time, scaled SegMamba did not obtain a significant improvement in accuracy compared to baseline SegMamba. It also required more GPU memory and achieved lower mDSC and higher mHD95 than Mamba-HoME.
>
> > W2: No user studies or clinical validation are provided.
>
> While we agree that user studies and clinical validation are important for eventual deployment, they are beyond the scope of this work. Our current work focuses on establishing the methodological foundation for reliable medical image segmentation and evaluating the proposed model under quantitative benchmarks, such as generalizability to cross-institutional (with 40 institutions for the in-house and 2 additional institutions for the remaining CT scans), multi-modality (CT, MRI, US) datasets and robustness to domain shift. We consider clinical validation an important direction for future work and are actively exploring clinical collaborations to that end.
>
> > W3: Some claims about “unprecedented memory and compute efficiency” may be overstated since inference speed lags behind simpler baselines.
>
> To avoid any overstatements, we will revise those sentences in the final version of the paper.
>
> > W4: Details about dataset preprocessing and training schedules are scattered across appendix sections.
>
> We appreciate the Reviewer’s feedback and agree that clarity in presenting dataset preprocessing and training details is essential. To improve readability, we have revised the Supplementary Material to consolidate all preprocessing steps and training schedules into a single dedicated section. We also added cross-references from the main paper for easier navigation. This improved Supplementary Material will be incorporated into the final version of the paper. We hope this improves the accessibility and reproducibility of our work.
>
> > Q1: Some sentences in Introduction are very long—could you break them up for clarity?
>
> We made sure to break up the very long sentences in the introduction section for clarity. For example, the sentence:
> “However, combining the global efficiency of SSMs with the localized adaptability of MoE remains largely unexplored, particularly in 3D medical imaging, where balancing efficiency and generalization across multi-modal datasets under resource constraints is paramount.”,
> was edited into:
> “However, combining the global efficiency of SSMs with the localized adaptability of MoE remains largely unexplored. Such a combination would particularly benefit 3D medical imaging, where balancing efficiency and generalization across multi-modal datasets under resource constraints is paramount.”
> These and other changes will be incorporated in the final version of the paper.
>
> > Q2: Please standardize font sizes across all tables for consistency?
>
> The final version will use standardized font sizes across all tables to ensure consistency.
>
> > Q3: How much improvement does HoME provide over plain Mamba when controlling for parameter count?
>
> Thank you for this excellent suggestion. To demonstrate the improvement over plain Mamba while controlling the parameter count, we compared to the plain Mamba-based segmentation method, SegMamba [1], with the number of parameters increased to approximately 176.5M, similar to that of Mamba-HoME. To this end, we increased:
> - the number of heads from [2, 2, 2, 2] to [4, 8, 16, 32],
> - the feature size from [48, 96, 192, 384] to [64, 128, 256, 512], and
> - the hidden size from 768 to 1024.
>
> This choice of SegMamba scaling was based on [2].
>
> Despite increasing the number of parameters, we did not observe improvements in segmentation performance by SegMamba (SegMamba-L in the Table below) compared to the baseline SegMamba (SegMamba-B). Notably, Mamba-HoME achieved a consistently superior mDSC and mHD95 compared to both baseline and scaled SegMamba.
>
> | Method | Parameters (M) | PANORAMA | MVSeg | AMOS-CT |
> |:--------------:|:--------------:|:------------:|:-------------:|:-------------:|
> |  SegMamba-B | 66.8  | 76.0 (14.1)  |  83.8 (15.6)  |  86.0 (98.9)  |
> |  SegMamba-L |  176.5  | 76.3 (11.8)  |  84.0 (14.8)  |  85.4 (96.2)  |
> |  Mamba-HoME |  170.5 | 77.5 (6.9)   |  84.8 (12.6)  |  86.3 (45.0)  |
>
> Values are shown as mDSC (%) and mHD95 (mm) in brackets.
>
> > Q4: What is the runtime and GPU memory footprint on a standard high-resolution volume?
>
> Table 1 presents the average runtime and GPU memory footprint on CT volumes, including high-resolution data, such as PANORAMA and our in-house datasets. Refer to Supplementary Material A.2 for dataset details.
>
> To further demonstrate performance on high-resolution cases, we performed additional experiments and selected the 100 cases with the highest resolution from the AbdomenAtlas 1.1 [3] dataset to report their specific runtime and GPU memory usage. Each CT scan consists of between 786 and 1,440 slices, with a mean of 1,156. The mean voxel spatial resolution is 0.82 × 0.82 × 0.79 mm³.
> For those very high-resolution data, Mamba-HoME achieves a good balance between inference efficiency and memory usage, with an inference time of 17.7 s and a moderate GPU memory footprint of 15.1 GB, outperforming other large-scale models in memory efficiency while maintaining competitive inference speed. Please note that the inference time includes data pre-processing, such as resampling the data.
>
> | Method | Parameters (M) | Inference time (s) | GPU memory (GB) |
> |------------------|:------------------:|:----------------------:|:------------------:|
> | uC 3DU-Net        |       21.7         |         13.4           |        17.5        |
> | SuPreM            |       62.2         |         19.1           |        21.0        |
> | Swin SMT          |      170.8         |         15.2           |        19.7        |
> | Swin UNETR        |       72.8         |         19.2           |        21.1        |
> | Hermes            |       44.5         |         21.5           |        17.1        |
> | VSmTrans          |       47.6         |         19.9           |        14.9        |
> | SegMamba-B        |       66.8         |         14.5           |        14.0        |
> | SegMamba-L        |      176.5         |         21.5           |        17.1        |
> | Mamba-HoME        |      170.5         |         17.7           |        15.1        |
>
> > Q5: Did you test cross-institution generalization or domain shift robustness?
>
> We did evaluate cross-institution generalization and robustness to domain shift, particularly for pancreas and PDAC segmentation. Specifically, we test Mamba-HoME on three external test sets: MSD Pancreas, NIH, and an in-house dataset. The in-house data includes samples from 40 independent medical centers (see Table 11 in the Supplementary Material). We also evaluated the performance of our model on various data domains, including CT, MRI and US. These benchmarks demonstrated excellent generalizability of Mamba-HoME to datasets across institutions and robustness to different data domains (see Tables 1-4).
>
> > Q6: Is there evidence that the second-level global routing meaningfully improves performance?
>
> Following this insightful suggestion, we performed an ablation study with three setups: (#1) Baseline - SegMamba, (#2) Mamba-HoME with first-level (local), and (#3) full Mamba-HoME with both first- and second-level routing. In this experiment, we used the FeTA 2022 dataset. This experiment evidenced that both first- and second-level routing indeed meaningfully improved the performance. This new result will be incorporated in the final version.
>
> | Setup (#) | mDSC (%) | mHD95 (mm) |
> |---------------------------|:------------:|:--------------:|
> | 1| 85.9 | 3.5 |
> | 2| 86.8 | 2.7 |
> | 3| 87.5 | 2.1 |
>
> > Q7: Are the observed improvements statistically significant on all datasets?
>
> We used the Wilcoxon signed-rank test to evaluate the statistical significance of performance differences between our proposed method, Mamba-HoME, and other state-of-the-art approaches. We observed statistically significant improvements across all methods and datasets (Wilcoxon rank-sum test; p < 0.05).
> We regret that this information was omitted from the main paper due to oversight and thank the reviewer for bringing this to our attention. This information will be added in the final version of the paper.
>
> **References**
>
> [1] Xing et al. "Segmamba: Long-range sequential modeling mamba for 3d medical image segmentation." MICCAI, 2024
>
> [2] Hatamizadeh et al. "Mambavision: A hybrid mamba-transformer vision backbone." CVPR, 2025
>
> [3] Li et al. "Abdomenatlas: A large-scale, detailed-annotated, & multi-center dataset for efficient transfer learning and open algorithmic benchmarking." Medical Image Analysis, 2024

---

> ### Author Response · Authors · 2025-08-08
> **Follow up on our rebuttal**
>
> Dear Reviewer,
>
> As less than one day remains before the Author–Reviewer discussion period closes, we would greatly appreciate it if you could take a look at our rebuttal to confirm whether we have addressed all of your concerns. In such a case, we would be grateful if you could consider raising your score.
>
> Thank you again for your time and thoughtful contributions to the process.
>
> Best regards,
>
> Authors

---

### Note · Authors · 2025-08-13

We thank the Area Chair for handling our manuscript and the Reviewers for their careful reading and constructive feedback, which have helped to further improve the quality of our work.

**Summary of Rebuttal Contributions**

* Conducted additional ablations to quantify the effect of varying the number of experts and group sizes on computational efficiency during inference, especially on high-resolution datasets
* Conducted an additional experiment to demonstrate superior segmentation performance and computational efficiency while controlling for the model’s parameter count
* Conducted an additional experiment to demonstrate segmentation performance and computational efficiency while controlling for the model’s parameter count
* Addressed minor issues, such as text within the Introduction, and standardized the font size in the tables

**Reviewer Evaluations**
* All Reviewers acknowledged our novel integration of SSMs with hierarchical two-level token routing, which effectively addresses the challenge of efficient local-to-global feature modeling in 3D medical image segmentation
* All Reviewers highlighted that experiments across five diverse datasets consistently showed improvement and appreciated the ablations demonstrating the impact of design and hyperparameter choices
* Reviewers PbeP and k1E5 recognized a well-executed methodology
* Reviewer SpNP appreciated “... the inclusion of pretraining and cross-modal transfer further strengthens the claim of generalizability.”

**Significance of this work**

Mamba-HoME tackles the challenge in 3D medical image segmentation of combining globally efficient long-range modeling with adaptive handling of diverse local patterns, by:
* Integrating SSMs with a hierarchical Soft Mixture-of-Experts (SMoE) for the first time in 3D medical image segmentation
* Introducing a two-level token-routing design: (1) Local SMoE routes grouped tokens to specialized local experts for fine-grained pattern learning, (2) Global SMoE aggregates outputs from local experts to capture comprehensive global context
* Designing a unified architectural block that merges memory-efficient long-sequence processing with hierarchical expert routing for scalable modeling
* Achieving state-of-the-art segmentation accuracy and generalization across five datasets, three modalities, and varying image resolutions


Overall, we trust that our work will serve as a valuable contribution to the NeurIPS community and to the field of AI in medical imaging.

---

### Decision · Program_Chairs · 2025-09-17

**Decision:**

Accept (poster)

**Comment:**

This paper presents a method for 3D medical image segmentation. Overall, the presentation is clear and easy to follow. The results show some improvement on 3D segmentation tasks. There are some concerns on the novelty, but the reviewers are more on the positive side of the work. Based on my reading of the rebuttals, the responses have addressed the concern of the comments, with confirmation from reviewers as well.